# Multiple dynamic interactions from basal ganglia direct and indirect pathways mediate action selection

Hao Li[1], Xin Jin[1,2,3]*

[1]Molecular Neurobiology Laboratory, The Salk Institute for Biological Studies, La Jolla, United States; [2]Center for Motor Control and Disease, Key Laboratory of Brain Functional Genomics, East China Normal University, Shanghai, China; [3]NYU–ECNU Institute of Brain and Cognitive Science, New York University Shanghai, Shanghai, China

**Abstract** The basal ganglia are known to be essential for action selection. However, the functional role of basal ganglia direct and indirect pathways in action selection remains unresolved. Here, by employing cell-type-specific neuronal recording and manipulation in mice trained in a choice task, we demonstrate that multiple dynamic interactions from the direct and indirect pathways control the action selection. While the direct pathway regulates the behavioral choice in a linear manner, the indirect pathway exerts a nonlinear inverted-U-shaped control over action selection, depending on the inputs and the network state. We propose a new center (direct)-surround (indirect)-context (indirect) 'Triple-control' functional model of basal ganglia, which can replicate the physiological and behavioral experimental observations that cannot be simply explained by either the traditional 'Go/No-go' or more recent 'Co-activation' model. These findings have important implications on understanding the basal ganglia circuitry and action selection in health and disease.

*For correspondence:
xjin@bio.ecnu.edu.cn

Competing interest: The authors declare that no competing interests exist.

## eLife assessment

In this **valuable** manuscript Li & Jin record from the substantia nigra and dorsal striatum to identify subpopulations of neurons with activity that reflects different dynamics during action selection, and then use optogenetics in transgenic mice to selectively inhibit or excite D1- and D2- expressing spiny projection neurons in the striatum, demonstrating a causal role for each in action selection in an opposing manner. They provide **solid** evidence for the argument that their findings cannot be explained by current models and propose a new 'triple control' model instead, with one direct and two indirect pathways, although direct evidence for a second indirect pathway is still lacking. These findings will be of broad interest to neuroscientists across multiple subfields.

## Introduction

Selecting the proper actions is essential for organism's survival and reproduction in the ever-changing environment (*Gallistel, 1980*). Numerous studies have implicated that the basal ganglia, a series of interconnected subcortical nuclei including the striatum and substantia nigra, play a primary role in action selection (*Graybiel, 1998*; *Hikosaka et al., 1998*; *Jin and Costa, 2015*; *Mink, 2003*; *Redgrave et al., 1999*). Indeed, a wide range of neurological and psychiatric disorders associated with the dysfunctional basal ganglia circuitry, including Parkinson's disease (*Benecke et al., 1987*), Huntington's disease (*Phillips et al., 1995*), obsessive-compulsive disorder (*Graybiel and Rauch, 2000*), are characterized by major deficits in action selection and movement control. Anatomically, commands for

motor control are processed by basal ganglia through two major pathways, termed direct and indirect pathway, originating from striatal D1- and D2-expressing spiny projection neurons (D1-/D2-SPNs), respectively (*Albin et al., 1989*; *DeLong, 1990*). These two pathways collectively modulate substantia nigra pars reticulata (SNr) activity and the basal ganglia output, thus influence behavioral decisions. There are currently two major types of thinking on how the basal ganglia pathways work. An early classic theory has suggested that the basal ganglia direct and indirect pathways oppose each other to facilitate and inhibit action, respectively (the 'Go/No-go' model) (*Albin et al., 1989*; *DeLong, 1990*; *Kravitz et al., 2010*). In contrast, a recent theory has proposed that direct pathway selects the desired action, while the indirect pathway inhibits other competing actions in order to highlight the targeted choice (the 'Co-activation' model) (*Cui et al., 2013*; *Hikosaka et al., 2000*; *Mink, 1996*).

The two theories have essentially agreed upon the function of direct pathway being the positive driving force for initiating or facilitating the desired actions. Yet, the ideas about the indirect pathway function are largely controversial as either impeding the desired action in the 'Go/No-go' model or inhibiting the competing actions in the 'Co-activation' model. While the precise neuroanatomy on how the D2-SPNs control SNr through indirect pathway has yet to be mapped out at single-cell level to differentiate the two hypotheses, either theory has found its supports from behavioral and physiological observations. For instance, it has been found that stimulation of striatal direct and indirect pathways can bidirectionally regulate locomotion (*Durieux et al., 2012*; *Kravitz et al., 2010*), consistent with the traditional 'Go/No-go' model. On the other hand, in vivo electrophysiological and imaging experiments revealed that the striatal direct and indirect pathways are both activated during action initiation (*Barbera et al., 2016*; *Cui et al., 2013*; *Geddes et al., 2018*; *Isomura et al., 2013*; *Jin et al., 2014*; *Klaus and Plenz, 2016*; *Markowitz et al., 2018*; *Nonomura et al., 2018*), as the 'Co-activation' model predicted. Furthermore, physiological and optogenetic studies concerning complex behavior such as learned action sequences have further complicated the issue, and unveiled various neuronal subpopulations in both pathways are activated during the initiation, termination, and switching of actions (*Geddes et al., 2018*; *Jin and Costa, 2015*; *Jin et al., 2014*; *Tecuapetla et al., 2016*). So far, how exactly the basal ganglia direct and indirect pathways work together to control action selection has been controversial and inconclusive, and the underlying circuit mechanism remains largely unclear (*Calabresi et al., 2014*).

Here, we trained mice to perform an operant action selection task where they were required to select one out of two actions to achieve reward, based on self-monitored time intervals (*Howard et al., 2017*). By employing in vivo neuronal recording, we found that the net output of two opponent SNr neuron populations is predictive of the behavioral choices. Through identifying striatal pathway-specific neuronal activity with optogenetic tagging, we found that there are neuronal populations in either the direct or indirect pathway that are activated during selecting one action and suppressed during another. Optogenetic inhibition, as well as selective ablation of direct pathway, impairs action selection, and optogenetic excitation of direct pathway enhances current choice, confirming a role of direct pathway in facilitating desired actions. Furthermore, optogenetic inhibition of indirect pathway improves action selection and excitation of indirect pathway impairs behavioral choices, as predicted by the 'Go/No-go' model. However, selective ablation of indirect pathway impairs action selection, opposite from the behavioral effect of optogenetic inhibition and at odds with the 'Go/No-go' model, but consistent with the prediction from the 'Co-activation' model. To resolve these contradictions, we propose a new center (direct)-surround (indirect)-context (indirect) 'Triple-control' functional model of basal ganglia pathways, in which there are two interacting indirect pathway subcircuits exerting opposite controls over the basal ganglia output. The new model can reproduce the neuronal and behavioral experimental results that cannot be simply explained by either the 'Go/No-go' or the 'Co-activation' model. Further systematic analyses from this new model suggested that the direct and indirect pathways modulate behavioral outputs in a linear and nonlinear manner, respectively. Notably, in the new 'Triple-control' model, the direct and indirect pathways can work together to dynamically control action selection and operate in a manner similar to 'Go/No-go' or 'Co-activation' model, depending on the activity level and the network state. These results revise our current understanding on how the basal ganglia control actions, and have important implications for a wide range of movement and psychiatric diseases where the dynamic balance between the two pathways is compromised (*Albin et al., 1989*; *Benecke et al., 1987*; *Calabresi et al., 2014*; *DeLong, 1990*; *Graybiel and Rauch, 2000*; *Mink, 1996*; *Phillips et al., 1995*).

## Results

### Opponent SNr activities underlie action selection

To address the role of basal ganglia in action selection, we trained mice in a recently developed 2–8 s task in which they are required to choose the left versus right action based on self-monitored time intervals (*Howard et al., 2017*). Specifically, mice were put into an operant chamber with both left and right levers extended (*Figure 1A*, see Methods). For a given trial, both levers retract at trial initiation, and after either 2 s or 8 s (50% for each, randomly interleaved), both levers extend. The mouse has to judge the interval between lever retraction and extension as 2 s vs. 8 s and make a corresponding action choice by pressing the left vs. right lever, respectively (*Figure 1A*). The first lever press after lever extension was registered as the mouse's choice. The correct choice leads to sucrose delivery (10 µl) as reward, and any lever presses beyond the first press after lever extension yield no outcome. The animal only has one chance to select the correct choice and gets rewarded in a given trial. If the animal's very first press after levers extension is the wrong choice, then there's no reward, and the chance to get rewarded in this particular trial vanishes, or the trial is functionally 'terminated' although both levers still available to press. The animal has no second chance to correct its wrong choice by pressing the correct lever after the wrong choice. During the 2 s vs. 8 s waiting period with lever retraction, the levers are not physically accessible to the animal. Even the animal is trying to approach to the lever during lever retraction, but no lever press will be generated (see *Video 1*). A new trial starts at lever retraction again after a random inter-trial-interval (ITI, 30 s on average; *Figure 1A*). Across 14 consecutive days of training, mice (*n*=10) significantly increased the correct rate of choice from chance level to more than 90% (*Figure 1B*). In addition, the animals gradually shortened the choice latency and demonstrated a strong preference toward the left lever due to its association with the shorter waiting time (*Figure 1—figure supplement 1A, B*). As a result, during the longer-waiting 8 s trials the mouse initially moved toward the left lever, then crossing the midpoint between left and right levers at around 4 s, and stayed around the right lever afterward (*Howard et al., 2017*; *Figure 1C*; *Video 1*). Note that, the mouse showed no stereotyped movement trajectories during the incorrect trials (*Figure 1C*). This emerged stereotyped movement trajectory in the 8 s trials thus provided us a unique opportunity for investigating the neural mechanisms underlying the internally driven, dynamic action selection process.

The SNr is one of the major output nuclei of basal ganglia (*Albin et al., 1989*; *DeLong, 1990*; *Hikosaka et al., 2000*; *Mink, 1996*). To investigate how the basal ganglia contribute to the dynamic process of action selection, we began by recording the SNr neuronal activity in mice trained in the 2–8 s task (*Figure 1D*, *Figure 1—figure supplement 1C*, see Methods). It was found that a large proportion (211/261, 80.8%; recorded from *n*=9 mice) of SNr neurons changed firing rate significantly during the correct 8 s trials as mice dynamically shifted the internal action selection from the left to the right (*Figure 1E*). The Z-score of the task-related neuronal firing rate, reflecting the firing activity changes related to baseline, was defined as firing rate index (FRI, see Methods). We focus on the data analyses in the 8 s trials since the first 2 s of 8 s trials consists of the identical behavioral and neuronal profiles of the 2 s trials due to the task design (*Figure 1C*, *Figure 1—figure supplement 1D–F*). The task-related SNr neurons were categorized into four subtypes based on the dynamics of FRI in the correct 8 s trials: Type 1 - monotonic decrease (*Figure 1E and F*, 102/211, 48.3%), Type 2 - monotonic increase (*Figure 1E and G*, 56/211, 26.5%), Type 3 - transient phasic increase (*Figure 1E and H*, 25/211, 11.9%), and Type 4 - transient phasic decrease (*Figure 1E,I*, 28/211, 13.3%). These four types of neuronal dynamics in SNr only appeared in the correct but not the incorrect trials (*Figure 1F–I*), nor on the day 1 of task training (*Figure 1—figure supplement 1G–K*), suggesting a tight correlation between the SNr neuronal dynamics and the behavioral performance. Here, we show trial-by-trial firing activities of SNr example neurons in correct 8 s trials from well-trained animals as follows. Although the time of initial approach to the left side varies across trials, trial-by-trial analysis showed that the firing activities are consistent across trials and the averaged activities faithfully reflect the dynamics of each trial, evident for all four types of neurons (*Figure 1—figure supplement 2A–D*). Specifically, the Type 1 and Type 2, but not the Type 3 and Type 4 neurons, exhibit firing changes co-varying with the action selection and these two types together consist in around 80% of all task-related SNr neuron population (*Figure 1J*, *Figure 1—figure supplement 2A–D*). There is no dramatic difference in dynamic subtypes and proportion between SNr neurons recorded in left and right hemispheres (*Figure 1—figure supplement 3*). Notably, for Type 1 neurons, the firing activities are much higher

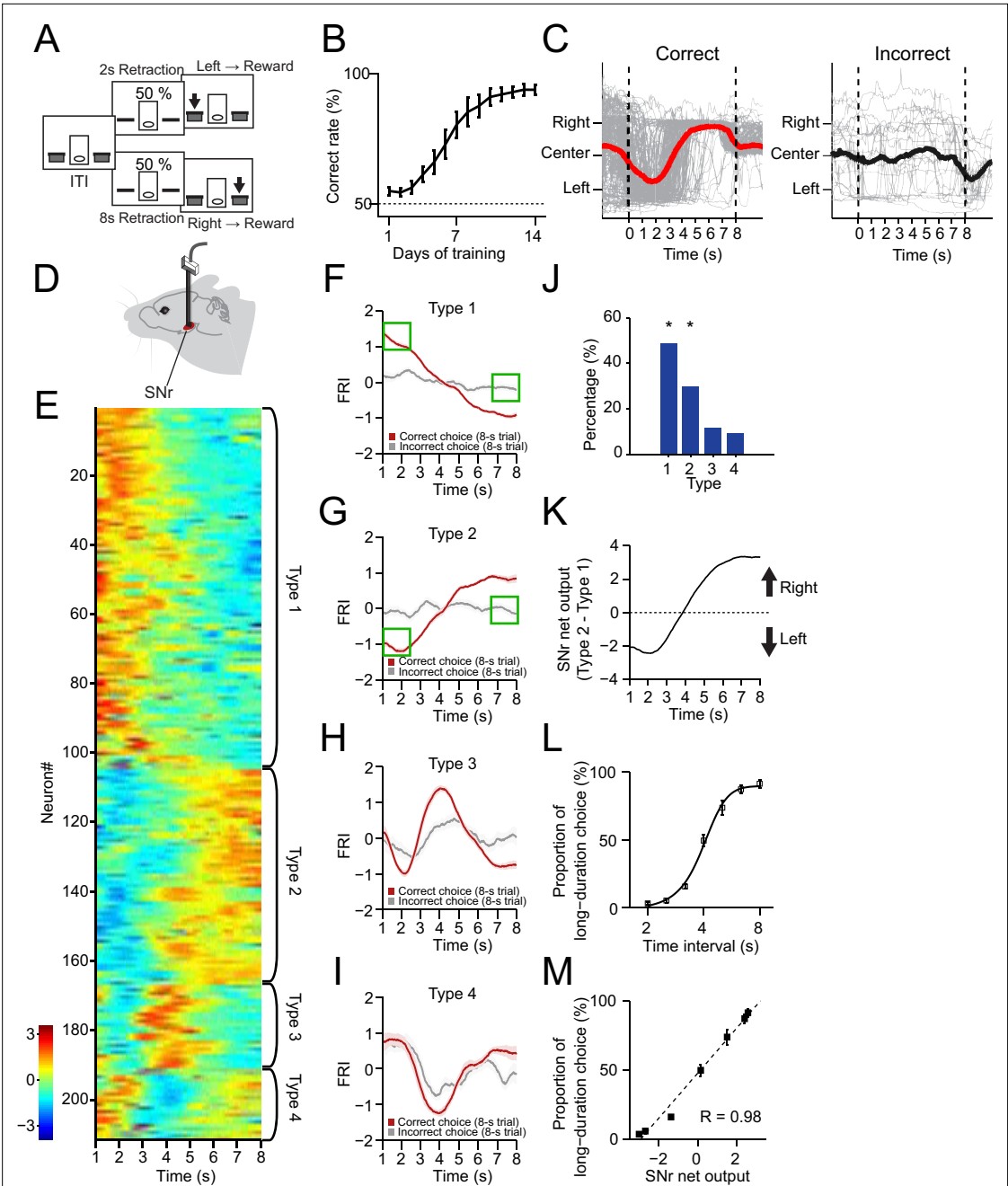

**Figure 1.** The neuronal dynamics in substantia nigra pars reticulata (SNr) during the 2–8 s action selection task. (**A**) Schematic diagram for the design of 2–8 s task. (**B**) Correct rate for wild-type mice across 14 days' training ($n$=10 mice, one-way repeated-measures ANOVA, significant effect of training days, $F_{13,117}$ = 32.54, p<0.0001). (**C**) Movement trajectory of an example mouse in correct (left panel) and incorrect (right panel) 8 s trials (gray line: trajectory of each trials; red/black line: the average trajectory). (**D**) Diagram of electrode array implanted into SNr. (**E**) Firing rate index (FRI) of neuronal activity for all task-related SNr neurons in correct 8 s trials. The magnitude of FRI is color coded and the SNr neurons are classified as four different types based on the activity dynamics. (**F–I**) Averaged FRI for Type 1 (F, green squares indicating activities related to left choice), Type 2 (G, green squares indicating activities related to left choice), Type 3 (**H**), Type 4 (**I**) of SNr neurons in correct (red) and incorrect 8 s trials (gray). (**J**) The proportion of four types of SNr neurons. Type 1 and Type 2 are major types and significantly more than Type 3 and Type 4 (Z-test, p<0.05). (**K**) SNr net output defined as the subtraction of averaged FRI between Type 1 and Type 2 SNr neurons. (**L**) Averaged psychometric curve ($n$=10 mice) of choice behavior. (**M**) The correlation between the Type 1 and Type 2 FRI subtraction and the behavioral choice ($R$=0.98, p<0.0005). Error bars denote s.e.m., same for below unless stated otherwise.

The online version of this article includes the following figure supplement(s) for figure 1:

**Figure supplement 1.** Behavioral performance across 14 days of training and the substantia nigra pars reticulata (SNr) neuronal recording on day 1.

*Figure 1 continued on next page*

*Figure 1 continued*

**Figure supplement 2.** Examples of substantia nigra pars reticulata (SNr) neuron and spiny projection neuron (SPN) subtypes.

**Figure supplement 3.** Substantia nigra pars reticulata (SNr) neuron activities in left and right hemisphere.

as animals selected left side at the correct 8 s trials than the firing activities when animals selected left side at the incorrect 8 s trials (*Figure 1F*, green squares). The same for Type 2 neurons, their firing activities are dramatically different when animals selected left side in the correct and incorrect trials (*Figure 1G*, green squares). Therefore, Type 1 and Type 2 dynamics cannot be simply explained by sensory or position-related neural activity. Furthermore, we compare the SNr neuron responses at rewarded and non-rewarded lever presses. For Type 1 SNr neurons, the firing activity at the rewarded left lever presses (defined as the left lever press in correct 2 s trials) is much higher than the firing activity at the non-rewarded left lever presses (defined as left lever presses in incorrect 8 s trials and random left lever press during the ITI). The firing activity difference can also be observed between the rewarded and non-rewarded right lever presses in Type 1 SNr neuron (*Figure 1—figure supplement 1L*). For Type 2 SNr neurons, although there's no difference between the rewarded and non-rewarded left lever presses, the firing activity at the rewarded right lever presses is higher than the firing activity at the non-rewarded right lever presses (*Figure 1—figure supplement 1L*). Again, given the same sensory inputs and spatial location for both rewarded and non-rewarded left presses, the difference between rewarded and non-rewarded lever presses indicates that the neural dynamics are action selection dependent, and not simply related to sensory or position information.

It has been suggested that SNr suppresses movements through the inhibition of downstream motor nuclei and releases action via disinhibition (*Hikosaka et al., 2000*; *Mink, 1996*). We thus ask whether the opponent neuronal dynamics in Type 1 and Type 2 SNr subpopulations mediate the dynamic shift of choice, by suppressing the competing selection of right vs. left action, respectively. Indeed, the SNr net output by subtracting Type 2 and Type 1 SNr neuronal dynamics (*Figure 1K*) is highly reminiscent of the animal's stereotyped movement trajectory during choice (*Figure 1C*). To further determine the relationship between the SNr net output and action selection, we tested the behavioral choice of the 2–8 s trained mice in a series of non-rewarded probe trials with novel intervals of 2.5 s, 3.2 s, 4 s, 5 s, and 6.3 s (see Methods). Consistent with what reported before (*Howard et al., 2017*), the probability of mice selecting the action associated with the long duration (8 s) gradually increases along with the time intervals of probe trials (*Figure 1L*). The resulting psychometric curve thus represents the animal's real-time action selection process during the 8 s trials. Further comparison between the psychometric curve and the SNr net output revealed a strong linear correlation (*Figure 1M*), indicating that the SNr net output faithfully predicts momentary behavioral choice. Together, these results suggest that mice can learn to dynamically shift their choice based on internally monitored time, and the opponent neuronal activities in SNr correlate with the action selection.

## SNr neuronal dynamics reflect action selection but not simply time or value

In the 2–8 s task, the passage of time and expectation of reward both change simultaneously with the animal's internal choice. One may argue that the Type 1 and Type 2 neuronal dynamics observed in SNr during the 8 s trials might reflect the passage of time or value of expected reward rather than action selection. To differentiate these possibilities and specify the functional role of SNr activity, we presented mice previously trained in the 2–8 s task with random probe trials of 16 s interval (*Figure 2A*). In these 16 s probe trials which they have never experienced before during training, the animals sometimes wait on the right side and press the right lever, or shift back to the left side and press the left lever when the levers are extended at 16 s (*Figure 2B and C*). This arbitrary choice situation in the 16 s probe trials thus

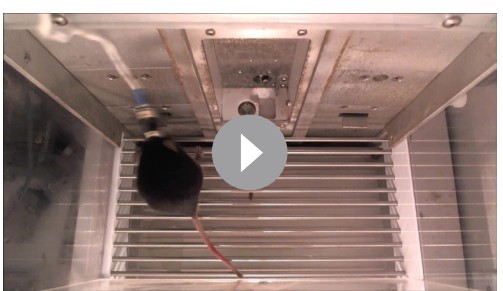

**Video 1.** Stereotypical behavior of a well-trained mouse in a successful 8 s trial.

https://elifesciences.org/articles/87644/figures#video1

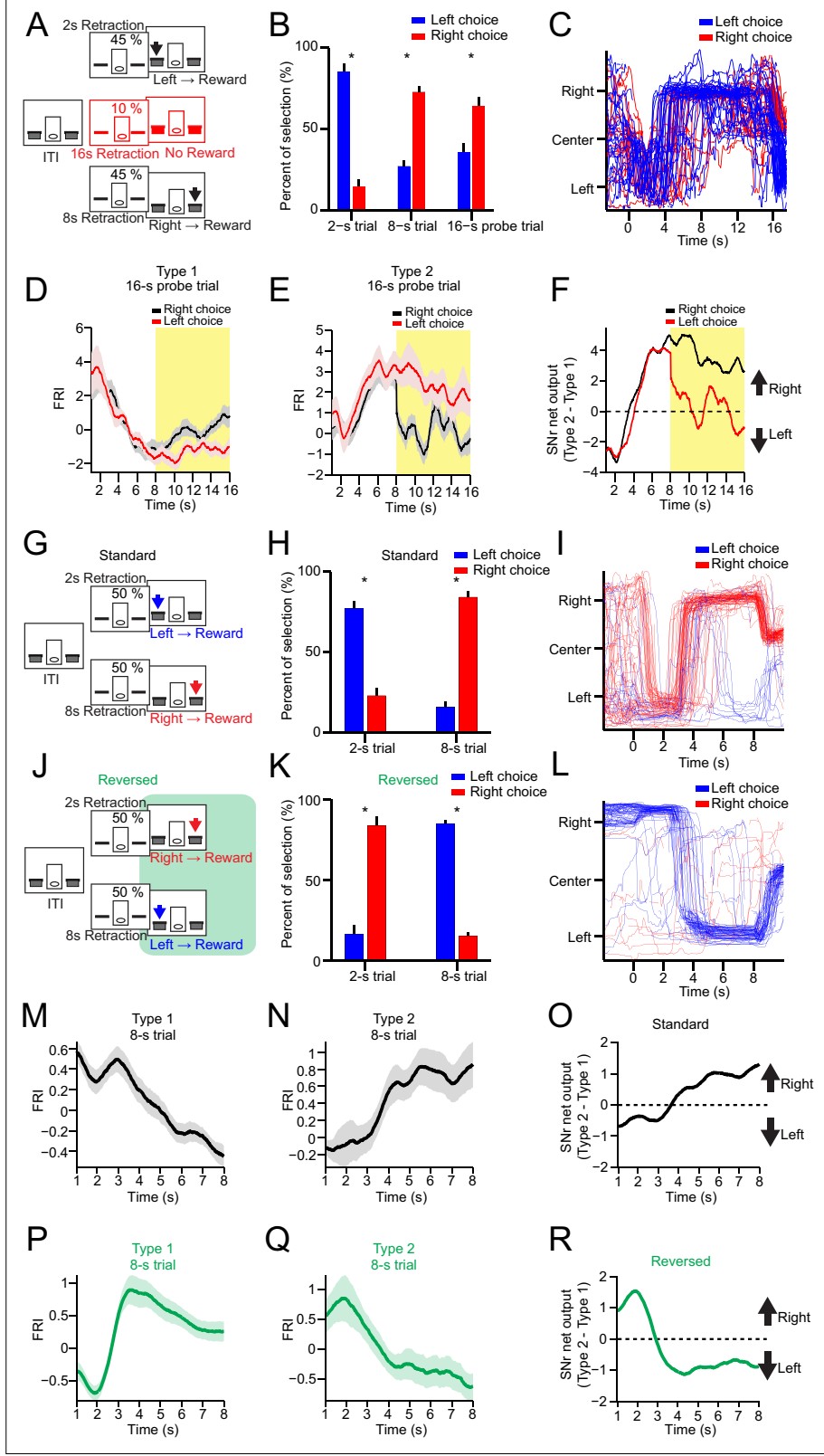

**Figure 2.** Substantia nigra pars reticulata (SNr) neuronal dynamics reflect action selection but not interval time or reward value. (**A**) Task diagram of 2–8 s control task with 10% 16 s probe trials. (**B**) Percentage of behavioral choice in 2 s, 8 s, and 16 s trials (blue: left choice; red: right choice) (*n*=9 mice, paired t-test, p<0.05). (**C**) Movement trajectory of an example mouse in 16 s trials (blue: left choice; red: right choice). (**D**) Averaged SNr Type 1 firing

*Figure 2 continued on next page*

*Figure 2 continued*

rate index (FRI) in 16 s trials (red: left choice; black: right choice). Firing rates from 8 s to 16 s (highlighted area) are compared between left and right choice (*n*=26 neurons, two-way repeated-measures ANOVA, significant difference between left and right choices, $F_{1,25}$=6.646, p=0.016). (**E**) Averaged SNr Type 2 FRI in 16 s trials (red: left choice; black: right choice). Firing rates from 8 s to 16 s are compared between left and right choice (*n*=16 neurons, two-way repeated-measures ANOVA, significant difference between left and right choices, $F_{1,15}$=5.785, p=0.029). (**F**) Subtraction of FRI for SNr Type 1 and Type 2 neurons in 16 s probe trials (red: left choice; black: right choice). (**G**) Task design of 2–8 s standard task. (**H**) Percentage of behavioral choice in 2 s and 8 s trials (blue: left choice; red: right choice) (*n*=6 mice, paired t-test, p<0.05). (**I**) Movement trajectory of an example mouse in 8 s trials (blue: left choice; red: right choice). (**J**) Task design of reversed 2–8 s task. (**K**) Percentage of behavioral choice in 2 s and 8 s trials in the reversed 2–8 s task (blue: left choice; red: right choice) (*n*=6 mice, paired t-test, p<0.05). (**L**) Movement trajectory of the same mouse as (**I**) in 8 s trials in the reversed 2–8 s task (blue: left choice; red: right choice). (**M**) Averaged FRI of the SNr Type 1 neurons in correct 8 s trials (*n*=14 neurons). (**N**) Averaged FRI of the SNr Type 2 neurons in correct 8 s trials (*n*=11 neurons). (**O**) SNr net output as the subtraction of FRI for SNr Type 1 (**M**) and Type 2 neurons (**N**) in the standard 2–8 s task. (**P**) Averaged FRI of the same neurons as (**M**) in correct 8 s trials of the reversed 2–8 s task. (**Q**) Averaged FRI of the same neurons as (**N**) in correct 8 s trials of the reversed 2–8 s task. (**R**) SNr net output as the subtraction of FRI for SNr Type 1 (**P**) and Type 2 neurons (**Q**) in the reversed 2–8 s task.

The online version of this article includes the following figure supplement(s) for figure 2:

**Figure supplement 1.** Behavioral statistics and neuronal dynamics of substantia nigra pars reticulata (SNr) neurons in the standard and reversed 2–8 s tasks.

provides a special window to determine the functional relationship between SNr activity and behavioral choice. If the Type 1 and Type 2 SNr subpopulations encode information about time passage or expectation value, their neuronal activities would continue changing monotonically between 8 s and 16 s. In contrast, if the Type 1 and Type 2 SNr subpopulations encode action selection, their neuronal activities would predict the behavioral choice and differentiate between the right vs. left action selection. Indeed, it was found that when the firing activity of Type 1 SNr neurons maintained below baseline from 8 s to 16 s, the mice tended to select the right lever later (*Figure 2D*). However, when the firing activity reversed the decreasing tendency to increase, the mice chose the left lever instead (*Figure 2D*). A similar relationship between the neuronal activity and behavior choice was also evident in Type 2 SNr neurons, albeit with opposite dynamics (*Figure 2E*). This is especially evident in the subtraction of Type 2 and Type 1 SNr neuronal dynamics, in which the SNr net output is strongly correlated with and predictive of behavioral choice (*Figure 2F*). These results thus suggested that the neuronal activities in SNr likely encode the ongoing action selection but not simply reflect time passage or reward value.

To further confirm this point, we recorded the firing activity from the same SNr neurons during both the 2–8 s control task (*Figure 2G*, 2 s-left and 8 s-right) and a modified version of 2–8 s task in which the contingency between action and interval is reversed (*Figure 2J*, 2 s-right and 8 s-left) on the same day (see Methods). It was found that the mice performed at around 80% correct in both tasks on the same day (*Figure 2H and K*, *Figure 2—figure supplement 1A*). Accordingly, the movement trajectories of the same mice in 8 s trials were reversed from left-then-right in the control task (*Figure 2I*) to right-then-left in the reversed task (*Figure 2L*). The left lever preference during the ITI in the control task was also switched to right lever preference in the reversed 2–8 s task (*Figure 2—figure supplement 1B*). Notably, the passage of time and expected value as well as other environmental factors are all identical in both versions of task, except that the animal's choice is now reversed from right to left for the 8 s trials (*Figure 2H, I* vs. *Figure 2K and L*). If Type 1 or Type 2 SNr neurons encode time or value, either neuronal population will exhibit the same neuronal dynamics in 8 s trials for both versions of task. On the other hand, if Type 1 and Type 2 SNr neurons encode action selection, their neuronal dynamics will reverse in the reversed version of 2–8 s task compared to the standard version. In fact, the Type 1 SNr neurons which showed monotonic decreasing dynamics in the control 2–8 s task (*Figure 2M*) reversed their neuronal dynamics to a monotonic increase in the reversed 2–8 s task (*Figure 2P*), consistent with the behavioral choice. The same reversal of neuronal dynamics was also observed in Type 2 SNr neurons in the reversed version of standard task (*Figure 2N and Q*). The SNr net output by subtracting Type 2 and Type 1 SNr neuronal dynamics, which was tightly correlated with the action selection in the standard 2–8 s task (*Figure 2O*), is reversed and now

predictive of the new behavioral choice in the reversed 2–8 s task (*Figure 2R*). Notably, Type 3 and Type 4 SNr neurons exhibiting transient change when mice switching between choices maintained the same neuronal dynamics in both tasks (*Figure 2—figure supplement 1C–F*). Together, these results therefore demonstrate that the output of basal ganglia reflects the dynamic action selection rather than simply time or value.

## Distinct striatal direct vs. indirect pathway activity during action selection

The basal ganglia output is largely controlled by two major neural pathways, called 'direct' and 'indirect' pathway, originating from D1- vs. D2-SPNs in the striatum, respectively (*Albin et al., 1989*; *DeLong, 1990*; *Hikosaka et al., 2000*; *Mink, 1996*). We then decided to determine the neuronal dynamics in the striatum, specifically the neuronal activity in the direct and indirect pathways during action selection. We employed in vivo extracellular electrophysiology to record the neuronal activity in the dorsal striatum when mice perform the 2–8 s task, and classified putative SPNs based on the spike waveforms and firing properties (*Geddes et al., 2018*; *Jin and Costa, 2010*; *Jin et al., 2014*). Among all the SPNs recorded from the trained mice (*n*=19), 341 out of 409 SPNs (83.4%) were defined as task-related neurons for showing significant firing changes during the 2 s and 8 s lever retraction period (*Figure 3A*, *Figure 1—figure supplement 2E–H*, *Figure 3—figure supplement 1A*). Similar to the various types of neuronal dynamics observed in SNr, task-related SPNs showed Type 1 (*Figure 3B*, monotonic decrease, 159/341, 46.6%), Type 2 (*Figure 3C*, monotonic increase, 103/341, 30.2%), Type 3 (*Figure 3D*, transient phasic increase, 49/341, 14.4%), and Type 4 (*Figure 3E*, transient phasic decrease, 30/341, 8.8%) activity profiles during the correct 8 s trials (*Figure 3A*, *Figure 1—figure supplement 2E–H*, *Figure 3—figure supplement 1A*). These neural dynamics were largely absent in SPNs on day 1 of training (*Figure 3—figure supplement 1B–F*). Also, SPNs recorded from left and right hemispheres showed similar proportions (*Figure 3—figure supplement 2*). These results indicate that the striatum, as one of the major input nuclei of basal ganglia, demonstrates the four types of neuronal dynamics similar with SNr during the dynamic process of action selection.

To further determine the neuronal activity in the direct and indirect pathways during action selection, we utilized an optogenetics-aided photo-tagging method (*Geddes et al., 2018*; *Howard et al., 2017*; *Jin and Costa, 2010*; *Jin et al., 2014*; *Lima et al., 2009*) to record and identify striatal D1- vs. D2-SPNs in freely behaving mice. Channelrhodopsin-2 (ChR2) was selectively expressed in D1- or D2-SPNs by injecting AAV-FLEX-ChR2 in the dorsal striatum of D1- and A2a-Cre mice, respectively (*Geddes et al., 2018*; *Jin et al., 2014*). In the end of each behavioral session with recording, optogenetic stimulation via an optic fiber attached to the electrode array was delivered to identify D1- vs. D2-SPNs through photo-tagging (*Figure 3F*, *Figure 3—figure supplement 1G–J*; *Geddes et al., 2018*; *Jin et al., 2014*). Only those neurons exhibiting a very short latency (≤6 ms) to light stimulation (*Figure 3G–I*) and showing identical spike waveforms (*R*≥0.95, Pearson's correlation coefficient) between behavior and light-evoked response (*Figure 3J and K*) were identified as Cre-positive thus D1- or D2-SPNs (*Geddes et al., 2018*; *Jin et al., 2014*). Within all positively identified D1-SPNs (*n*=92 from 6 mice) and D2-SPNs (*n*=95 from 6 mice), 74 out of 92 (80.4%) D1-SPNs and 79 out of 95 (83.1%) D2-SPNs showed a significant change in firing rate during the correct 8 s trials. In addition, all four types of neuronal dynamics during action selection were found in both D1-SPNs (*Figure 3L and M*) and D2-SPNs (*Figure 3N and O*), as observed in SNr. The Type 1 and Type 2 neuronal dynamics showing monotonic firing change (*Figure 3M and O*) were the predominant task-related subpopulations within either D1- (*Figure 3L*) or D2-SPNs (*Figure 3N*). Notably, the striatal D1-SPNs consist of significantly more Type 1 than Type 2 neurons (*Figure 3L*), while D2-SPNs show a similar proportion between the two types (*Figure 3N*). These data thus suggest while neurons in both the striatal direct and indirect pathways encode information related to behavioral choice, the two pathways might reflect and contribute to distinct aspects of action selection.

## Ablation of striatal direct vs. indirect pathway differently impaired action selection

Given the action selection-related neuronal dynamics observed in striatum, we next asked whether the neural activity in striatum is necessary for learning and execution of action selection, and furthermore, what is the functional difference between the direct and indirect pathways. It has been reported

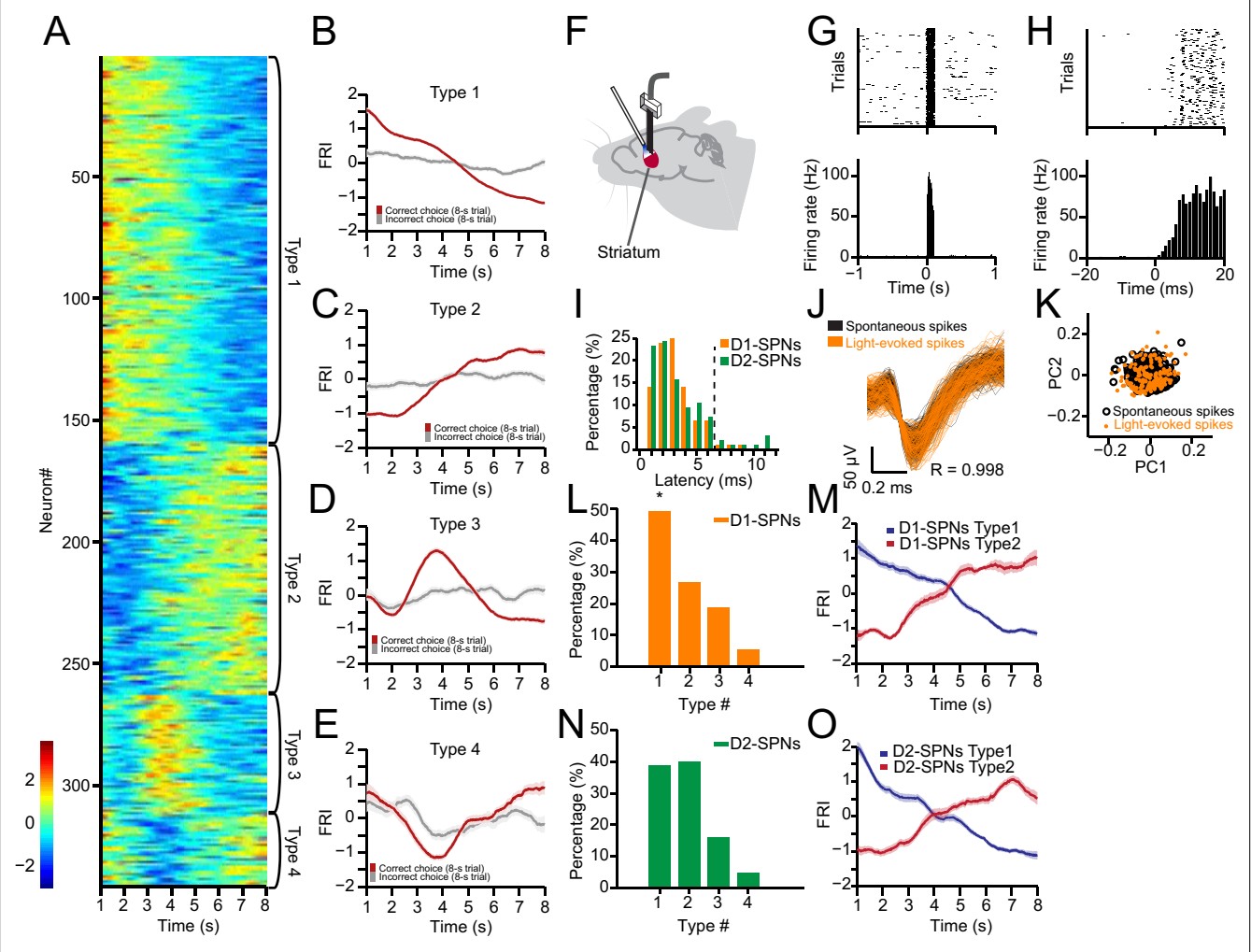

**Figure 3.** Neuronal activity of striatal D1- and D2-expressing spiny projection neurons (D1- and D2-SPNs) during action selection. (**A**) Firing rate index (FRI) of neuronal activity for all task-related SPNs in correct 8 s trials. SPNs were classified as Types 1–4. (**B–E**) Averaged FRI for Type 1 (**B**), Type 2 (**C**), Type 3 (**D**), Type 4 (**E**) of SPNs in correct (red) and incorrect 8 s trials (gray). (**F**) Diagram of simultaneous neuronal recording and optogenetic identification of D1- vs. D2-SPNs in dorsal striatum. (**G**) Top panel: Raster plot for a representative D1-SPN response to 100 ms optogenetic stimulation. Each row represents one trial and each black dot represents a spike. Bottom panel: Peristimulus time histogram (PETH) aligned to light onset at time zero. (**H**) PETH for the same neuron as shown in (**G**) with a finer time scale. (**I**) Distribution of light response latencies for D1- and D2-SPNs. (**J**) Action potential waveforms of the same neuron in (**G**) for spontaneous (black) and light-evoked (orange) spikes ($R$=0.998, p<0.0001, Pearson's correlation). (**K**) Principal component analysis (PCA) of action potential waveforms showing the overlapped clusters of spontaneous (black) and light-evoked (orange) spikes. (**L**) Proportion of D1-SPN subtypes. Type 1 neurons are significantly more than other three types of neurons in D1-SPNs (Z-test, p<0.05). (**M**) Averaged FRI for Type 1 (blue) and Type 2 (red) D1-SPNs in correct 8 s trials. (**N**) Proportion of D2-SPN subtypes. (**O**) Averaged FRI for Type 1 (blue) and Type 2 (red) D2-SPNs in correct 8 s trials.

The online version of this article includes the following figure supplement(s) for figure 3:

**Figure supplement 1.** Striatum neuronal recording on day 1 of training, recording array, and optic fiber placement validation.

**Figure supplement 2.** Striatal projection neuron activities in left and right hemisphere.

that the NMDA receptors on striatal SPNs are critical for sequence learning (*Geddes et al., 2018*; *Jin and Costa, 2010*) and action selection (*Howard et al., 2017*). To further identify the functional role of NMDA receptors on D1- vs. D2-SPNs for action selection, we employed a genetic strategy to specifically delete NMDA receptors from D1- vs. D2-SPNs by crossing mice carrying a floxed NMDAR1 (NR1) allele with a dorsal striatum-dominant D1-cre line (*Gong et al., 2007*) and A2a-cre line (*Geddes et al., 2018*; *Jin et al., 2014*), respectively (referred to as D1-NR1 KO and D2-NR1 KO mice, respectively; see Methods). Both the D1-NR1 KO and D2-NR1 KO mice are significantly impaired in learning the 2–8 s task compared to their littermate controls (*Figure 4A and B*), suggesting that NMDA receptors

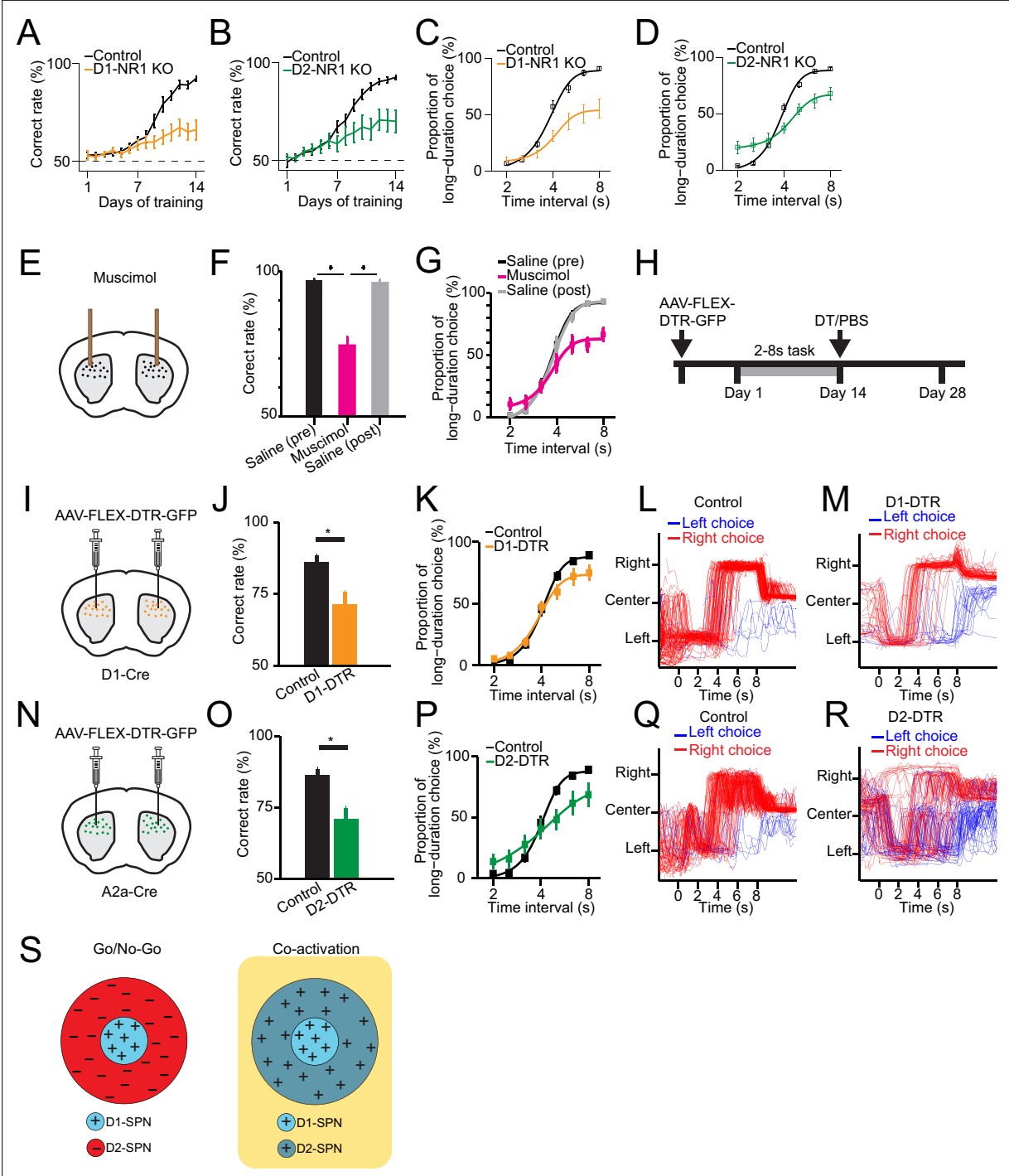

**Figure 4.** Selective genetic knockout and ablation of D1- or D2-expressing spiny projection neurons (D1- or D2-SPNs) distinctly alters action selection. (**A**) Correct rate of control (*n*=11 mice) and D1-NR1 KO mice (*n*=16) in 2–8 s task during 14 days' training (two-way repeated-measures ANOVA, significant difference between control and KO mice, $F_{1,25}$=10.8, p=0.003). (**B**) Correct rate of control (*n*=17) and D2-NR1 KO mice (*n*=10) in 2–8 s task during 14 days' training (two-way repeated-measures ANOVA, significant difference between control and KO mice, $F_{1,25}$=8.728, p=0.007). (**C**) The psychometric curve for control (*n*=11) and D1-NR1 KO mice (*n*=16) (two-way repeated-measures ANOVA, significant difference between control and KO mice, $F_{1,25}$=12.27, p=0.002). (**D**) The psychometric curve for control (*n*=17) and D2-NR1 KO mice (*n*=10) (two-way repeated-measures ANOVA, significant difference between control and KO mice, $F_{1,25}$=9.64, p=0.005). (**E**) Schematic of muscimol infusion into the dorsal striatum in trained mice. (**F**) Correct rate for control (black: pre-muscimol, gray: post-muscimol) and mice with muscimol infusion (magenta) in dorsal striatum (*n*=9 mice, paired t-test, p<0.01). (**G**) The psychometric curve for control (*n*=9 mice, black: pre-muscimol, gray: post-muscimol control) and mice with muscimol infusion (*n*=9 mice, magenta) in dorsal striatum (two-way repeated-measures ANOVA, significant difference between control and muscimol infusion, $F_{2,16}$=11.74,

*Figure 4 continued on next page*

*Figure 4 continued*

p=0.0007). (**H**) Timeline for selective diphtheria toxin (DT) ablation experiments. (**I**) Schematic of diphtheria toxin receptor (DTR) virus (AAV-FLEX-DTR-GFP) injection in dorsal striatum of D1-Cre mice. (**J**) Correct rate for control (*n*=9 mice) and mice with dorsal striatum D1-SPNs ablation (D1-DTR, *n*=8 mice) (two-sample t-test, p=0.0016). (**K**) The psychometric curve for control (*n*=9 mice) and D1-SPNs ablation mice (*n*=8 mice) (two-way repeated-measures ANOVA, main effect of ablation, $F_{1,15}$=1.84, p=0.195; interaction between trial intervals and ablation, $F_{6,90}$=4.14, p=0.001). (**L**) Movement trajectory of a control mouse in 8 s trials. (**M**) Movement trajectory of a D1-DTR mouse in 8 s trials. (**N**) Schematic of DTR virus (AAV-FLEX-DTR-GFP) injection in dorsal striatum of A2a-Cre mice. (**O**) Correct rate for control (*n*=8 mice) and mice with dorsal striatum D2-SPNs ablation (D2-DTR, *n*=8 mice) (two-sample t-test, p=0.005). (**P**) The psychometric curve for control (*n*=9 mice) and D2-SPNs ablation mice (*n*=8 mice) (two-way repeated-measures ANOVA, main effect of ablation, $F_{1,15}$=0.477, p=0.5; interaction between trial intervals and ablation, $F_{6,90}$=12.6, p<0.001). (**Q**) Movement trajectory of a control mouse in 8 s trials. (**R**) Movement trajectory of a D2-DTR mouse in 8 s trials. (**S**) Schematic of center-surround receptive field diagram for Go/No-Go (left) and Co-activation (right) models. '+' indicates facilitating effect to selection. '-' indicates inhibitory effect to selection.

The online version of this article includes the following figure supplement(s) for figure 4:

**Figure supplement 1.** Simulation of lesion experiments in Go/No-Go, Co-activation, and combination models.

on either D1- or D2-SPNs are critical for learning of proper action selection. In the end of 2-week training, when given the probe trials with various intervals across 2–8 s, it was found that D1-NR1 KO mice showed a systematic bias toward the lever associated with short interval and made deficient behavioral choice only in long interval trials (*Figure 4C*). In contrast, D2-NR1 KO mice showed impaired action selection across various probe trials of both short and long intervals (*Figure 4D*). These data suggest that while NMDA receptors on both D1- and D2-SPNs are required for action learning, the deletion of NMDA receptors in direct and indirect pathways impairs action selection in a different manner.

We then asked whether that neural activity in dorsal striatum is necessary for the proper execution of action selection after learning. We first conduct striatal inactivation in trained wild-type mice by bilateral intra-striatal infusion of muscimol (*Figure 4E*, see Methods). Striatal muscimol infusion significantly reduced the animal's overall performance in comparison with the pre- and post-saline injection controls (*Figure 4F*). When tested with probe trials, the psychometric curve indicated that the inactivation of striatum impairs action selection for the long trials (*Figure 4G*). These data thus suggested that the striatal neural activity is critical for appropriate execution of learned action selection.

To further elucidate the functional role of specific striatal pathways in action selection, we next employed a viral approach to bilaterally express diphtheria toxin receptors (DTR) (AAV-FLEX-DTR-eGFP) in the dorsal striatum of trained D1- and A2a-Cre mice, followed by diphtheria toxin (DT) injections to selectively ablate D1- or D2-SPNs (*Geddes et al., 2018*; *Figure 4H, I, and N*; see Methods). Ablation of either D1- or D2-SPNs significantly impaired action selection and reduced the correct rate of choice (*Figure 4J and O*). Notably, the psychometric curve revealed that D1-SPNs ablation mice showed a selective impairment of choice in long interval trials (*Figure 4K*). In contrast, mice with D2-SPNs ablation exhibited choice deficits in both long and short trials (*Figure 4P*). Consistent with the D1- and D2-NR1 KO data, these results suggest that the direct and indirect pathways are both needed yet play distinct roles in action selection.

The classic 'Go/No-go' model of basal ganglia suggests the direct and indirect pathways work antagonistically to release and inhibit action, respectively (*Albin et al., 1989*; *DeLong, 1990*; *Kravitz et al., 2010*). On the other hand, more recent 'Co-activation' model of basal ganglia proposes that direct pathway initiates the selected action and at the same time, the indirect pathway inhibits the competing actions (*Cui et al., 2013*; *Hikosaka et al., 2000*; *Mink, 1996*). For visualization purpose, we diagram 'Go/No-go' and 'Co-activation' models as center-surround receptive field with D1-SPNs as the center and D2-SPNs as the surround (*Figure 4S*; *Figure 4—figure supplement 1A, D*). The 'center-surround' layout is derived from the receptive field of neurons in the early visual system, as an intuitive analogy in describing the functional interaction among striatal pathways (*Mink, 2003*). The area of each region does not represent the amount of cells but mainly qualitative functional role (*Figure 4S*). While the direct pathway plays the similar role in both models (*Figure 4—figure supplement 1B, E*), the function of indirect pathway differs dramatically (*Figure 4S*). Lesion of the indirect pathway thus leads to contrast predictions on action selection from the two models (*Figure 4—figure supplement 1C, F*). Specifically, ablation of D2-SPNs would facilitate the action being selected through removing inhibition according to the Go/No-go model (*Figure 4—figure supplement 1C*; *Albin et al., 1989*; *DeLong, 1990*; *Kravitz et al., 2010*), while blockage of indirect pathway would

impair the action selection due to disinhibition of competing actions according to the Co-activation model (*Figure 4—figure supplement 1F*; *Cui et al., 2013*; *Hikosaka et al., 2000*; *Mink, 1996*). Although our D1-SPNs ablation experiment indicates that direct pathway is required for action selection as suggested in both models (*Figure 4J and K*), the D2-SPNs ablation result favorably supports the Co-activation model over the Go/No-go model (*Figure 4O and P*, *Figure 4—figure supplement 1F*). In fact, close inspection of the movement trajectories of D1-SPNs lesioned mice in the 8 s trials showed that compared to control mice (*Figure 4L*), they tend to stick on the left side more often with impaired right choice when lever extension at 8 s (*Figure 4M*). In contrast, D2-SPNs lesioned mice demonstrated overall rather random movement trajectories, and the stereotyped left-then-right movement sequences were largely disrupted in comparison with the controls (*Figure 4Q and R*). These observations are mostly consistent with the idea of indirect pathway inhibiting competing actions in the Co-activation model (*Figure 4S*) and lesion of indirect pathway disrupts action selection for both the short and long trials (*Figure 4P–R*). Together, these data suggest that ablation of direct and indirect pathways both impair choice behavior but in a distinct manner due to their different roles in action selection.

## Optogenetic manipulation of D1- vs. D2-SPNs distinctly regulates action selection

To further determine the specific function of direct vs. indirect pathway in action selection, we employed optogenetics to alter the D1- and D2-SPNs activity in vivo with high temporal precision and investigated its effects on the ongoing action selection process. Both the classic 'Go/No-go' (*Albin et al., 1989*; *DeLong, 1990*; *Kravitz et al., 2010*) and more recent 'Co-activation' (*Cui et al., 2013*; *Hikosaka et al., 2000*; *Mink, 1996*) models predict that activation of the direct pathway enhances the action selection (*Figure 5—figure supplement 1A, E, I, K, O, Q*), while inhibition of direct pathway reduces the correct choice (*Figure 5—figure supplement 1B, F, L, R*). To experimentally validate the models' predictions, AAV-FLEX-ChR2 was injected into the dorsal striatum of D1- or A2a-Cre mice and optic fibers were implanted bilaterally for in vivo optogenetic stimulation (*Figure 5A*, *Figure 3—figure supplement 1K,L*; see Methods) (*Geddes et al., 2018*; *Jin et al., 2014*). After mice learned

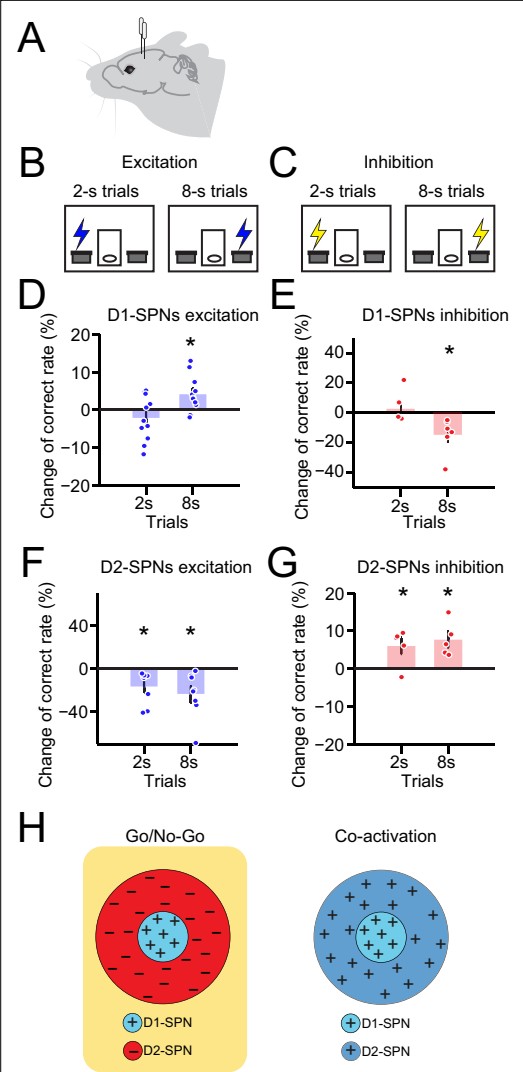

**Figure 5.** Optogenetic manipulation of D1- vs. D2-expressing spiny projection neurons (D1- vs. D2-SPNs) differently regulates action selection. (**A**) Schematic of optic fiber implantation for experimentally optogenetic excitation or inhibition of D1- or D2-SPNs in the dorsal striatum. (**B, C**) Schematic for optogenetic excitation (**B**) and inhibition (**C**) of D1-/D2-SPNs for 1 s right before lever extension in 2–8 s task. (**D**) Change of correct rate for optogenetic excitation of D1-SPNs in 2 s and 8 s trials (*n*=11 mice, one-sample t-test, 2 s trials: p=0.248; 8 s trials: p<0.05). (**E**) Change of correct rate for optogenetic inhibition of D1-SPNs in 2 s and 8 s trials (*n*=6 mice, one-sample t-test, 2 s trials: p=0.557; 8 s trials: p<0.05). (**F**) Change of correct rate for optogenetic excitation of D2-SPNs in 2 s and 8 s trials (*n*=8 mice, one-sample t-test, 2 s trials: p<0.05; 8 s trials: p<0.05). (**G**) Change of correct rate for optogenetic inhibition of D2-SPNs in 2 s and 8 s trials (*n*=5 mice, one-sample t-test, 2 s trials: p<0.05; 8 s trials: p<0.05). (**H**) Schematic of center-surround receptive field diagram for Go/No-Go (left) and Co-

*Figure 5 continued on next page*

*Figure 5 continued*

activation (right) models. '+' indicates facilitating effect to selection. '-' indicates inhibitory effect to selection.

The online version of this article includes the following figure supplement(s) for figure 5:

**Figure supplement 1.** Simulation of optogenetic manipulation in Go/No-Go and Co-activation models.

the 2–8 s task, 1 s pulse of constant light (wave length 473 nm) was delivered right before lever extension in randomly chosen 50% of 2 s and 50% of 8 s trials (*Figure 5B and C*, see Methods). The correct rate of optogenetic stimulation trials is compared with the non-stimulation trials of the same animal as a within-subject design. We observed no significant change on the correct rate in 2 s trials, whereas the correct rate was significantly increased by optogenetic stimulation in 8 s trials (*Figure 5D*), indicating a facilitation effect on action selection by stimulating the D1-SPNs. We then sought to determine the effect of suppressing D1-SPN activity on action selection by viral expression of Halorhodopsin (AAV5-EF1a-DIO-eNpHR3.0-eYFP) in the dorsal striatum of D1-cre mice (*Gradinaru et al., 2010*). As expected, inhibiting D1-SPNs right before lever extension in trained mice reduced the correct rates in 8 s but not 2 s trials (*Figure 5C and E*), opposite to D1-SPN stimulation effects. These experimental data with bidirectional optogenetic manipulation suggest that the D1-SPN activity is positively correlated with the choice performance, consistent with the hypothesis of direct pathway facilitating the action selected in both the Go/No-go and Co-activation models (*Figure 5—figure supplement 1K, L, Q, R*).

Nevertheless, the two models have distinct views on the function of indirect pathway. While the classic 'Go/No-go' model suggests that the indirect pathway inhibits the selected action (*Albin et al., 1989*; *DeLong, 1990*; *Kravitz et al., 2010*), the 'Co-activation' model hypothesizes that the indirect pathway inhibits the competing actions instead (*Cui et al., 2013*; *Hikosaka et al., 2000*; *Mink, 1996*). These models thus provide contrasting predictions about the effect of activation of the indirect pathway on action selection, being decreased correct rate based on the Go/No-go model (*Figure 5—figure supplement 1M*) and increased correct rate from the Co-activation model (*Figure 5—figure supplement 1S*), respectively. We thus decided to test the distinct predictions from the two models by optogenetic manipulation of indirect pathway during action selection in the 2–8 s task. ChR2 or Halorhodopsin (eNpHR3.0) was expressed in the dorsal striatum of A2a-cre mice for bilaterally optogenetic activation or inhibition during behavior (*Figure 5A*, *Figure 3—figure supplement 1L*; see Methods). Notably, optogenetic excitation of D2-SPNs for 1 s right before lever extension decreased the correct rate in both 2 s and 8 s trials (*Figure 5F*). In contrast, transient optogenetic inhibition of D2-SPNs before behavioral choice increased correct rates for both 2 s and 8 s trials (*Figure 5G*). These data suggest that opposite to the D1-SPN manipulation, optogenetic stimulation of D2-SPNs impairs action selection, while inhibition of D2-SPNs facilitates behavioral choice. These optogenetic results further unveil the distinct roles of direct vs. indirect pathway in action selection, and are in line with the predictions from the Go/No-go (*Figure 5H*, *Figure 5—figure supplement 1M, N*) but not the Co-activation model (*Figure 5—figure supplement 1S, T*).

## A 'Triple-control' model of basal ganglia circuit for action selection

Our DT lesion experiments found that ablation of indirect pathway impairs action selection (*Figure 4O and P*), as predicted from the Co-activation but not Go/No-go model (*Figure 4S*), while the optogenetic results suggested that inhibition of D2-SPNs enhances behavioral choice (*Figure 5G*), a result in favor of the Go/No-go rather than Co-activation model (*Figure 5H*). We wonder whether these seemly discrepant effects are attributed to a more complex circuit mechanism involving in the indirect pathway different from either the Go/No-go or Co-activation model. To systematically investigate the cell type- and pathway-specific mechanisms underlying action selection, we first add Go/No-go and Co-activation models together to examine whether the resulted combination model could explain the experimental observations (*Figure 4—figure supplement 1G*). The lesion of D1-SPNs in the combination model indeed selectively impaired choice in long interval trials (*Figure 4—figure supplement 1H*). However, the effect of D2-SPNs ablation in the combination model was neutralized due to the opposing contributions from Go/No-go and Co-activation models, respectively (*Figure 4—figure supplement 1I*). Based on these simulation results, none of the Go/No-go, Co-activation, and combination models was able to fully capture the underlying mechanism of basal ganglia in action selection. Inspired by the data in current experiments, we decided to build a new computational model of the

cortico-basal ganglia circuitry based on the realistic neuroanatomy (*Aoki et al., 2019*; *Mailly et al., 2003*; *Schmidt and Berke, 2017*; *Taverna et al., 2008*) and empirical neuronal physiology during action selection (*Figures 1–3*).

Different from the dual control of action by direct vs. indirect pathway in either the Go/No-go or Co-activation model (*Figure 4—figure supplement 1*), our new model adds an additional layer of control derived from the indirect pathway, thus called 'Triple-control' model for action selection. The combination of Go/No-go or Co-activation models clearly failed to explain all the experimental results (*Figure 4—figure supplement 1G–I*), therefore in our model, the new layer of control is not a simple add-on but equipped with interaction with other layers. Specifically, the new model consists of one direct pathway and two indirect pathways defined as D2-SPN #1 and D2-SPN #2 two subpopulations, corresponding to the Co-activation and Go/No-go functional modules, respectively (*Figure 6A and B*). In addition, the indirect pathway D2-SPNs in the Co-activation module inhibits the indirect pathway D2-SPNs in the Go/No-go module through the well-known D2-SPN collaterals with the properties of short-term depression (STD) in the striatum (*Burke et al., 2017*; *Gustafson et al., 2006*; *Schmidt and Berke, 2017*; *Taverna et al., 2008*; *Tecuapetla et al., 2007*; *Figure 6A*; see Methods), providing asymmetric modulation to D2-SPN subgroups and promoting Co-activation module as the dominant functional module at rest. In this 'Triple-control' basal ganglia model, striatal D1- and D2-SPNs associated with left and right actions receive excitatory inputs from corresponding cortical inputs (*Figure 6A*) to generate Type 1 and Type 2 neuronal dynamics (*Figure 6—figure supplement 1A–D*; *Lo and Wang, 2006*). The D1- and D2-SPNs then regulate the SNr neuronal dynamics through the direct and indirect pathways, respectively (*Figure 6—figure supplement 1*; *Albin et al., 1989*; *DeLong, 1990*; *Hikosaka et al., 2000*; *Mink, 1996*). The net SNr output (*Figure 6—figure supplement 1F, I*), which controls the downstream brainstem and thalamic circuits necessary for action selection (*Hikosaka, 2007*; *Lo and Wang, 2006*; *Redgrave et al., 1999*), will determine the final behavioral choice (*Figure 6—figure supplement 1G, J*). The choice preference toward left lever over right lever was reflected in the direct pathway by the unevenly weighted connection strength from cortex to D1-SPN Left/Right, as well as the connection strength from D1-SPN Left/Right to SNr Left/Right neurons (see Methods). Our computational simulations showed that this 'Triple-control' network model could faithfully recapitulate the neuronal activity across the basal ganglia circuitry and predict the behavioral choice (*Figure 6—figure supplement 1*).

To dissect the functional role of direct vs. indirect pathway in action selection, we simulate the cell ablation experiments and examine the behavioral output in the 'Triple-control' basal ganglia model. Ablation of D1-SPNs in the network model (*Figure 6—figure supplement 1E*) modulates both Type 1 and Type 2 SNr dynamics but in different magnitude due to the biased striatal input to SNr left output and mutual inhibition between SNr left vs. right outputs (*Figure 6—figure supplement 1F*; see Methods). As a result, the lesion causes a downward shift in the net SNr output, especially evident at the late section of 8 s (*Figure 6—figure supplement 1G*). This change in net SNr output predicts a behavioral bias toward left choice as seen in the psychometric curve (*Figure 6C*), consistent with experimental results in mice with D1-SPNs ablation (*Figure 4K*). In contrast, ablation of D2-SPNs in the network model (*Figure 6—figure supplement 1H*), by removing the indirect pathways of both the Go/No-go and Co-activation modules, alters Type 1 and Type 2 SNr dynamics (*Figure 6—figure supplement 1I*) and change the net SNr output dramatically around 2 s as well as 8 s (*Figure 6—figure supplement 1J*). The model thus predicts behavioral choice deficits for both short and long trials during D2-SPNs ablation (*Figure 6D*), consistent with experimental observations (*Figure 4P*). Together, these data suggest that our new 'Triple-control' basal ganglia model, based on realistic neuroanatomy and empirical neuronal physiology, can perform action selection similar to the behavior of mice, and successfully replicate the pathway-specific lesion effects on choice.

We further simulate the neuronal and behavioral effects of optogenetic manipulation of D1- and D2-SPNs in the cortico-basal ganglia model. Consistent with the experimental results (*Figure 5D and F*), optogenetic stimulation of D1-SPNs facilitates the ongoing choice (*Figure 6E*), while optogenetic inhibition of D1-SPNs suppresses ongoing choice in the model (*Figure 6G*). In addition, optogenetic stimulation of D2-SPNs impairs the ongoing choice and causes switching (*Figure 6F*), while optogenetic inhibition of D2-SPNs facilitates ongoing choice, due to the now dominant Go/No-go module mediated by the short-term depression of D2 collaterals in the model (*Figure 6H*). Consistent with

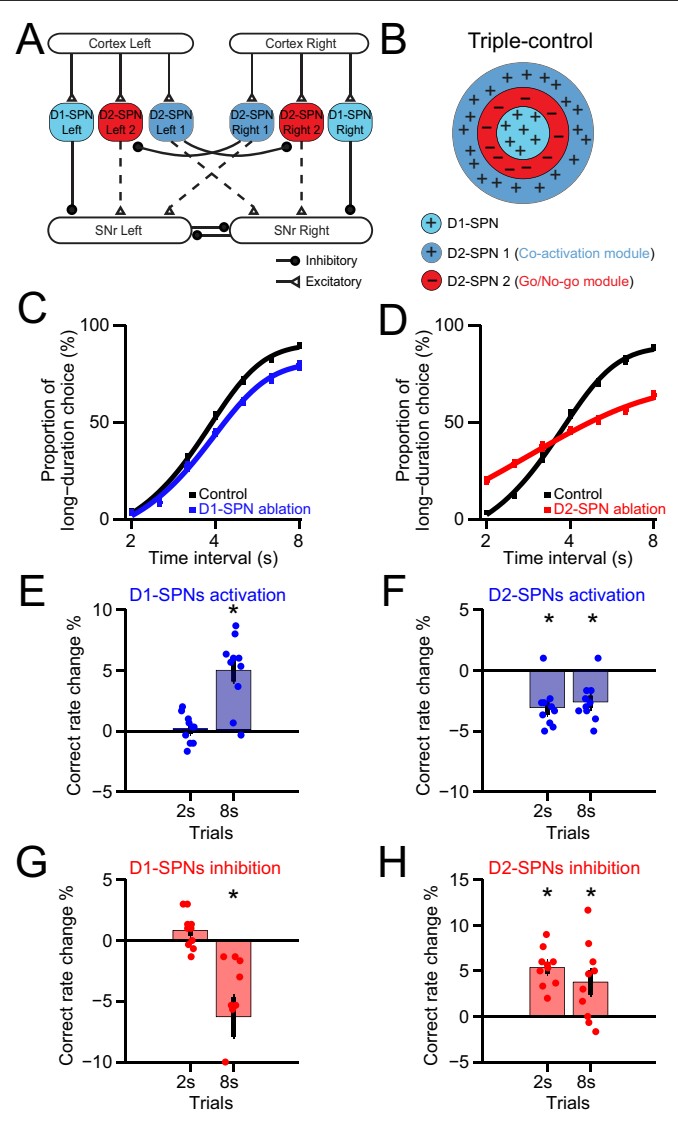

**Figure 6.** A Triple-control computational model of basal ganglia direct and indirect pathways for action selection. (**A**) Network structure of the cortico-basal ganglia model based on realistic anatomy and synaptic connectivity. Dashed lines indicate multi-synaptic connections. (**B**) Schematic of center-surround-context receptive field diagram for 'Triple-control' model. '+' indicates facilitating effect to selection. '-' indicates inhibitory effect to selection. (**C**) The psychometric curves of behavioral output in control (black) and D1-expressing spiny projection neurons (D1-SPNs) ablation condition (blue) in 'Triple-control' model ($n=10$, two-way repeated-measures ANOVA, main effect of ablation, $F_{1,18}=98.72$, $p<0.0001$; interaction between trial intervals and ablation, $F_{6,108}=7.799$, $p<0.0001$). (**D**) The psychometric curves of behavioral output in control (black) and D2-SPNs ablation condition (red) in 'Triple-control' model ($n=10$, two-way repeated-measures ANOVA, main effect of ablation, $F_{1,18}=99.54$, $p<0.0001$; interaction between trial intervals and ablation, $F_{6,108}=177.6$, $p<0.0001$). (**E**) Change of correct rate for optogenetic excitation of D1-SPNs in 2 s and 8 s trials ($n=10$, one-sample t-test, 2 s trials: $p=0.407$; 8 s trials: $p<0.05$). (**F**) Change of correct rate for optogenetic excitation of D2-SPNs in 2 s and 8 s trials ($n=10$, one-sample t-test, 2 s trials: $p<0.05$; 8 s trials: $p<0.05$). (**G**) Change of correct rate for optogenetic inhibition of D1-SPNs in 2 s and 8 s trials ($n=10$, one-sample t-test, 2 s trials: $p=0.28$; 8 s trials: $p<0.05$). (**H**) Change of correct rate for optogenetic inhibition of D2-SPNs in 2 s and 8 s trials ($n=10$, one-sample t-test, 2 s trials: $p<0.05$; 8 s trials: $p<0.05$).

The online version of this article includes the following figure supplement(s) for figure 6:

**Figure supplement 1.** The neuronal activities in the 'Triple-control' model and simulation of lesion experiments.

**Figure supplement 2.** Optogenetic activation of D1- vs. D2-expressing spiny projection neurons (D1- vs. D2-SPNs) differently regulates substantia nigra pars reticulata (SNr) activities in model and experiments.

*Figure 6 continued on next page*

*Figure 6 continued*

**Figure supplement 3.** Computational modeling of optogenetic manipulation reveals that D1- vs. D2-expressing spiny projection neurons (D1- vs. D2-SPNs) differently regulates substantia nigra pars reticulata (SNr) outputs in the 'Triple-control' model.

the experimental observations, the optogenetic inhibition effect is opposite from the D2-SPNs cell ablation in the model (*Figure 6D*).

We next investigate how the striatum influences SNr outputs in the model. Since the collateral projection with STD in D2-SPNs is the key in our 'Triple-control' model to switch between Go/No-go and Co-activation modules, we first built a motif of indirect pathway with two D2-SPNs subgroups defined as D2-SPN #1 and D2-SPN #2 (*Figure 6—figure supplement 2A*). We tested this indirect pathway motif with monotonic neural dynamics observed in experiments meanwhile simulating the optogenetic activation at 1 s and 7 s (*Figure 6—figure supplement 2D–I*). The SNr therefore received more activation at 1 s than at 7 s (*Figure 6—figure supplement 2J, K*), suggesting that the D2-SPNs with short-term depression in collateral inhibition modulates SNr activities in a firing rate-dependent manner.

We next sought to test the model's predictions and experimentally investigate the distinctions in modulating SNr activities between the direct and indirect pathways during action selection. In order to manipulate D1- or D2-SPNs and monitor SNr responses at the same time, we simultaneously implanted optogenetic fibers and recording array into striatum and SNr respectively on a single mouse (*Figure 6—figure supplement 2L*). While the mice performing the 2–8 s task, optogenetic stimulation was delivered to activate either D1- or D2-SPNs. It was found that optogenetic activation of D1- or D2-SPNs caused both inhibition and excitation in Type 1 and Type 2 SNr neurons (*Figure 6—figure supplement 2M–O*). To further compare SNr activities responding to striatal activation at different time points during the lever retraction period, for a given trial, we activated D1-SPNs (or D2-SPNs) either at 1 s or 7 s after the lever retraction (*Figure 6—figure supplement 2P–R*). For direct pathway, the change of FRI in SNr activities caused by activation of D1-SPNs showed no significant difference between 1 s and 7 s (*Figure 6—figure supplement 2P, S*). For indirect pathway, activating D2-SPNs at 1 s caused smaller activation of FRI than at 7 s in Type 1 SNr neurons (*Figure 6—figure supplement 2Q*), whereas for Type 2 SNr neurons, activating D2-SPNs at 1 s induced bigger FRI increase at 1 s than at 7 s (*Figure 6—figure supplement 2R*). Overall, activating D2-SPNs tended to bias the firing rate downward at 1 s but upward at 7 s in Type 1 SNr neurons, which was counteractive to the decreasing tendency of Type 1 SNr neuron (*Figure 6—figure supplement 2T*). In contrast, Type 2 SNr neurons showed higher FRI increase and smaller decrease in response to activating D2-SPNs at 1 s than at 7 s, which was opposing to the increasing dynamics of Type 2 SNr neurons (*Figure 6—figure supplement 2T*). This firing rate-dependent modulation on SNr activities through indirect pathway is consistent with the computational simulation (*Figure 6—figure supplement 2J, K*; *Figure 6—figure supplement 3*). Therefore, the underlying D2-SPNs collaterals might indeed be a key mechanism contributing to the modulation of SNr activity and action selection in vivo, as simulated in the 'Triple-control' model.

Taken together, our new 'Triple-control' basal ganglia model, based on realistic neuroanatomy and empirical neurophysiology, successfully reproduces both the lesion and optogenetic data we collected during the animal experiments. It could thus potentially provide essential insights into the circuit mechanism of basal ganglia underlying action selection.

## Linear and nonlinear control of action selection by direct vs. indirect pathway

To gain an overall picture of how basal ganglia control action selection, we run through the model with a wide continuous range of manipulation to mimic the effects from lesion to optogenetic inhibition and optogenetic activation (*Figure 7A, B, E, and F*). The simulations of cell ablation and bidirectional optogenetic manipulations of D1-SPNs activity in the model reveal no significant effects at 2 s trials (*Figure 7C*), but a linear relationship between the neuronal activity in direct pathway and the behavioral performance of choice in 8 s trials (*Figure 7D*), as observed in animal experiments. It thus further confirms that direct pathway selects action and facilitates ongoing choice, consistent with the

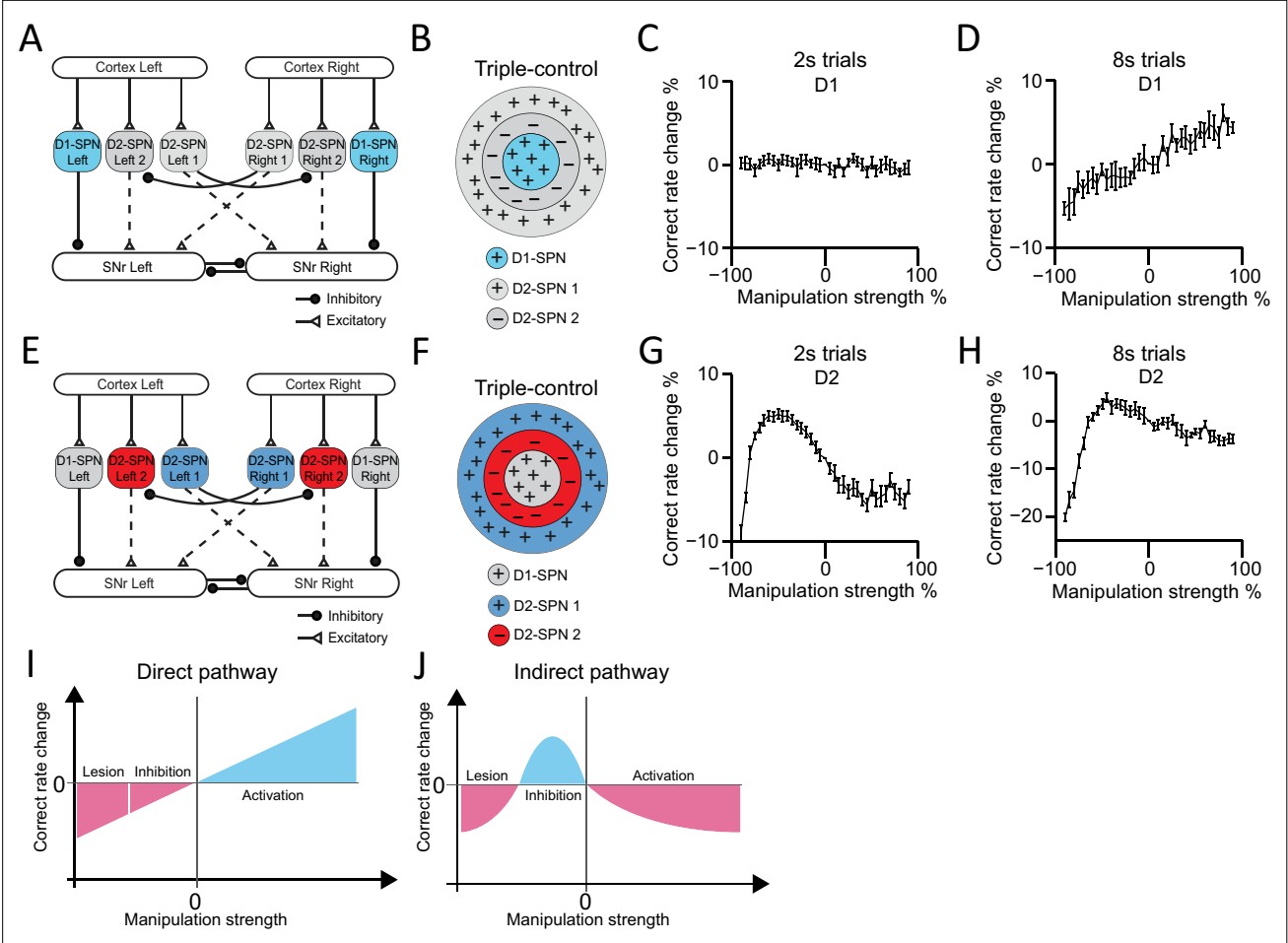

**Figure 7.** Computational modeling reveals direct and indirect pathways regulating action selection in a distinct manner. (**A**) Schematic for manipulation of D1-expressing spiny projection neurons (D1-SPNs) in 'Triple-control' model. (**B**) Schematic of manipulation of D1-SPNs in the center-surround-context receptive field diagram for 'Triple-control' model. '+' indicates facilitating effect to selection. '-' indicates inhibitory effect to selection. (**C**) Correct rate change in 2 s trials when manipulating D1-SPNs with different manipulation strengths (*n*=10, one-way repeated-measures ANOVA, effect of manipulation strength, *F*36,324=1.171, p=0.238). (**D**) Correct rate change in 8 s trials when manipulating D1-SPNs with different manipulation strengths (*n*=10, one-way repeated-measures ANOVA, effect of manipulation strength, *F*36,324=13.71, p<0.0001). (**E**) Schematic for optogenetic manipulation of D2-SPNs in 'Triple-control' model. (**F**) Schematic of manipulation of D2-SPNs in the center-surround-context receptive field diagram for 'Triple-control' model. '+' indicates facilitating effect to selection. '-' indicates inhibitory effect to selection. (**G**) Correct rate change in 2 s trials when manipulating D2-SPNs with different manipulation strengths (*n*=10, one-way repeated-measures ANOVA, effect of manipulation strength, *F*36,324=59.13, p<0.0001). (**H**) Correct rate change in 8 s trials when manipulating D2-SPNs with different manipulation strengths (*n*=10, one-way repeated-measures ANOVA, effect of manipulation strength, *F*36,324=40.75, p<0.0001). (**I**) Diagram of linear modulation of direct pathway. (**J**) Diagram of nonlinear modulation of indirect pathway.

The online version of this article includes the following figure supplement(s) for figure 7:

**Figure supplement 1.** Computational modeling of manipulation reveals that Go/No-Go and Co-activation model differently predicts the behavioral outcomes.

**Figure supplement 2.** Computational modeling reveals that the linear and nonlinear modulation of action selection by direct versus indirect pathway qualitatively hold with additional striatal collateral connections.

**Figure supplement 3.** Computational modeling of dopaminergic modulation in the 'Triple-control' model.

predictions from both the classic Go/No-go and recent Co-activation models (*Figure 7I*, *Figure 7—figure supplement 1A–D*; *Albin et al., 1989*; *DeLong, 1990*; *Hikosaka et al., 2000*; *Mink, 1996*).

In contrast, manipulations of D2-SPNs activity from cell ablation to optogenetic inhibition and then optogenetic stimulation in the model demonstrate an inverted-U-shaped nonlinear relationship between the neuronal activity in indirect pathway and action selection, for both 2 s and 8 s trials (*Figure 7G and H*). Detailed analyses reveal that D2-SPNs ablation removes both the Co-activation

and Go/No-go module in the indirect pathway and leaves SNr activity dictated by D1-SPN inputs. However, due to the inhibition from Co-activation to Go/No-go module in the indirect pathway via D2-SPN collaterals and short-term plasticity of these synapses (*Figure 6—figure supplement 2A–C*; see Methods), optogenetic manipulation of D2-SPNs differentially affects the D2-SPN subpopulations groups and promotes Go/No-go module to dominate the basal ganglia network (*Figure 6—figure supplement 2C–S*). This dynamic switch of dominance between Co-activation and Go/No-go modules on the basal ganglia network gives rise to a nonlinear relationship between D2-SPNs manipulation and the behavioral outcome (*Figure 7J*).

Note that when the same inputs were applied to the Go/No-go or Co-activation model alone, the behavioral performance in either model exhibits linear negative (*Figure 7—figure supplement 1E*) or positive correlation (*Figure 7—figure supplement 1F*) with D2-SPNs activity, respectively. Both our experimental and modeling results thus indicate that different from either the Go/No-go or Co-activation model, the indirect pathway regulates action selection in a nonlinear manner, depending on the state of the network and D2-SPNs activity level. Besides collaterals within D2-SPNs, other collateral connections, for example connections between D1-SPNs or connections between D1- and D2-SPNs, could also contribute to the regulation of action selection (*Taverna et al., 2008*). We tested our 'Triple-control' model with adding additional collateral connections as D1→D1 (*Figure 7—figure supplement 2A–C*), D1→D2 (*Figure 7—figure supplement 2D–F*), and D2→D1 (*Figure 7—figure supplement 2G–I*), respectively. It was found while these additional collaterals further quantitatively regulate action selection, the general principle of linear vs. nonlinear modulation of action selection by direct and indirect pathways still qualitatively hold (*Figure 7—figure supplement 2*). Interestingly, our current 'Triple-control' model can also replicate the behavioral effects of optogenetic manipulation of nigrostriatal dopamine on behavioral choice (*Howard et al., 2017*), and further unveils an inverted-U-shaped relationship between striatal dopamine concentration change and action selection (*Figure 7—figure supplement 3*). Together, these results suggest that there are multiple levels of interactions from D1- and D2-SPNs to dynamically control SNr output, and the basal ganglia direct and indirect pathways distinctly control action selection in a linear and nonlinear manner, respectively.

## Discussion

Here, by using an internally driven 2–8 s action selection task in mice, we investigated the function of basal ganglia direct and indirect pathways in mediating dynamic action selection. We found that the neuronal activities in SNr, the major output of basal ganglia, directly reflect animals' internal action selection process, other than simply time or value. It was also observed that the striatum, the main input of basal ganglia, shares the similar action selection-related neuronal dynamics with SNr and is needed for both learning and execution of proper action selection. Furthermore, the striatal direct and indirect pathways exhibit distinct neuronal activity and during manipulation, they have different functional effects on controlling action selection. Notably, the experimental observations on the physiology and function of direct and indirect pathways cannot be simply explained by either the traditional 'Go/No-go' model or the more recent 'Co-activation' model. We proposed a new 'Triple-control' functional model of basal ganglia, suggesting a critical role of dynamic interactions between different neuronal subpopulations within the indirect pathway for controlling basal ganglia output and behavior. In the model, a 'center (direct pathway)-surround (indirect pathway)-context (indirect pathway)' three layers of structure exerts dynamic control of action selection, depending on the input level and network state. This new model respects the realistic neuroanatomy, and can recapitulate and explain the essential in vivo electrophysiological and behavioral findings. It also provides a new perspective on understanding many behavioral phenomena involving in dopamine and basal ganglia circuitry in health and disease.

Our current 2–8 s action selection task offers a unique opportunity to observe the animal's internal switch from one choice to another and monitor the underlying neuronal dynamics correspondingly. We observed two major types of monotonically changing SNr neuronal dynamics during the internal choice switching, presumably one type associated with selecting one action and another with selecting the competing action, respectively. The classic view on SNr activity is that it tonically inhibits the downstream motor nuclei and releases action via disinhibition (*Albin et al., 1989*; *DeLong, 1990*; *Hikosaka and Wurtz, 1983*; *Wurtz and Hikosaka, 1986*). The increased response in SNr, however, could potentially inhibit the competing actions or the movements toward the opposing direction through

projections to the contralateral brain regions like superior colliculus (*Jiang et al., 2003*). Here, we found that two subpopulations in SNr showed opposite monotonic firing change during the left-then-right choice, and notably, their neuronal dynamics switched when the animals performed the reversed version of task which requires a right-then-left choice. It thus suggests that these SNr neurons are indeed associated with different action options during choice behavior, and actively adjust their firing rates to facilitate respective action selection. Given the opposite neuronal dynamics and functionally antagonistic nature of Type 1 and Type 2 SNr neurons, we defined the net output of basal ganglia by the subtracting the neuronal activity between the two SNr subpopulations and correlated it with the behavioral choice. The subtraction between Type 1 and Type 2 SNr neurons is the net output of two competing choices and indicates animals' choice in real time. Also, signals corresponding to left and right choices through direct/indirect pathways eventually converge to SNr (*Albin et al., 1989*; *DeLong, 1990*). The collateral inhibition within SNr (*Brown et al., 2014*; *Mailly et al., 2003*) gives rise to the direct competition between different SNr functional subgroups. Therefore, the subtraction between Type 1 and Type 2 SNr neurons represents the outcome of competition between choices. Indeed, we found that the basal ganglia net output exhibited a tight correlation with the psychometric curve of behavioral choice, and faithfully represented a neural basis for the dynamic action selection process.

As one of the major input nuclei of basal ganglia, striatum influences SNr activity through direct/indirect pathways and undisputedly, plays an essential role in action selection (*Ding and Gold, 2012*; *Geddes et al., 2018*; *Jin et al., 2014*; *Lauwereyns et al., 2002*; *Tai et al., 2012*). By genetic manipulation and pharmacological inactivation, we showed that striatum is indispensable for both learning action selection and the proper performance of learned behavioral choice. The recording of neuronal activity in dorsal-central striatum during action selection further revealed that striatal spiny projection neurons share the similar types of neuronal dynamics as SNr. Through optogenetic tagging in freely behaving mice, we further found that dorsal-central striatal SPNs in the direct and indirect pathways show distinct activity profile, with D1-SPNs representing a strong bias toward the preferred choice, while D2-SPNs encoding two choices equally.

Two prevailing models have been proposed to explain the functional distinction between D1- and D2-SPNs. The canonical model of the basal ganglia suggests that the D1- and D2-SPNs play antagonistic roles in controlling action as mediating 'Go' and 'No-go' signals, respectively (*Albin et al., 1989*; *DeLong, 1990*; *Kravitz et al., 2010*). A more recent model, however, implies that as D1-SPNs initiate an action, D2-SPNs co-activate with D1-SPNs to inhibit other competing actions (*Cui et al., 2013*; *Hikosaka et al., 2000*; *Isomura et al., 2013*; *Jin et al., 2014*; *Mink, 1996*). Essentially, these two models agree upon the functional role of D1-SPNs in releasing or facilitating the desired action, but contradict on the function of D2-SPNs on which targeted action of inhibiting. Here, our in vivo recording data indicate that both D1- and D2-SPNs share similar neuronal dynamics during action selection, and the neural activity alone is not sufficient to separate and determine whether 'Go/No-go' or 'Co-activation' model is supported (*Cui et al., 2013*; *Isomura et al., 2013*; *Jin et al., 2014*). To resolve the functional distinction of the direct vs. indirect pathway, we applied a series of cell-type-specific manipulations on striatal D1- and D2-SPNs during action selection behavior. First, we generated mutant mice in which NMDA receptors are deleted from either striatal D1- or D2-SPNs (*Geddes et al., 2018*; *Jin et al., 2014*). Both the D1-NR1 KO and D2-NR1 KO mice showed learning deficits and behavioral choice impairments when tested with probe trials, suggesting that both D1- and D2-SPNs are necessary for learning and performing action selection. Notably, while the D1-NR1 KO mice are mostly impaired in the choice associated with 8 s, a less-preferred option compared to 2 s, the D2-NR1 KO mice are compromised in both 2 s and 8 s choice. Additional experiments with cell-type-specific ablation further confirmed these results, consistent the distinct neuronal activity profile in these two pathways revealed during in vivo neuronal recording. While both the 'Go/No-go' and 'Co-activation' models predict the suppression of D1-SPNs activity leads to impaired action selection, supported by current KO and cell ablation data, the manipulation experiments on D2-SPNs favor the 'Co-activation' but not the 'Go/No-go' model which the latter suggests D2-SPNs ablation would improve rather than impair action selection.

Next, we directly introduced transient bidirectional manipulations to D1- and D2-SPNs activity by optogenetics while mice performing the task. Our findings revealed that activation or inhibition of D1-SPNs increased and decreased the correct rate of choice respectively, suggesting a facilitating role

of direct pathway in action selection, which again fits well with the 'Go/No-go' as well as the 'Co-activation' model. In contrast, optogenetic activation of D2-SPNs decreased the correct rate of choice, while inhibition of D2-SPNs promoted the correct choice. When stimulating D2-SPNs, animals are still able to press the lever and make a selection shortly after lever extension, therefore, the behavioral effect triggered by D2 stimulation is not simply due to a general effect of decreased locomotion, but the altered action selection process. These results were supportive to the 'Go/No-go' model but contradicted to the prediction of 'Co-activation' theory, which the latter predicts that activation of D2-SPNs inhibits competing actions to facilitate desired choice, whereas inhibition of D2-SPNs releases competing actions and compromises the ongoing choice.

In summary, neither 'Go/No-go' nor 'Co-activation' models could fully explain the experimental results we found, particularly for experiments on D2-SPNs in the indirect pathway. Through computer simulation, we further demonstrated that a simple additive combination of 'Go/No-go' nor 'Co-activation' models by linear addition cannot reproduce all the experimental observations. To resolve these theoretical difficulties, we proposed a new center-surround-context 'Triple-control' model of basal ganglia pathways for action selection. Specifically, two subpopulations of D2-SPNs in the indirect pathway function as 'Co-activation' and 'Go/No-go' modules respectively, and an activity-dependent inhibition from 'Co-activation' to 'Go/No-go' module mediates the dynamic switch between the dominant module depending on the inputs and network state. Due to the dominant 'Co-activation' module in the default state, excessive inhibition of D2-SPNs or ablating the entire indirect pathway eliminates the promotive contribution and impairs action selection in the 'Triple-control' model, consistent with the experimental observations. In contrast, transient increase of D2-SPNs firing activity during optogenetic stimulation introduces shift toward the 'Go/No-go' dominance from the 'Co-activation' module via firing rate-dependent short-term depression of the inhibitory synapses between them, which amplifies the 'No-go' signal and compromises action selection as experimentally found. In contrast, transient decrease of D2-SPNs firing activity during optogenetic inhibition results in disinhibition of 'Go/No-go' module from inhibitory control of 'Co-activation' module, with an attenuated 'No-go' signal which leads to better performance in choice. These results from our new 'Triple-control' model thus suggest that the basal ganglia circuitry could be much more dynamic than previously thought, and it could employ a complex mechanism of functional module reconfiguration for context- or state-dependent flexible control. More importantly, our model further proposed that while direct pathway regulates action selection in a linear manner, the indirect pathway modulates action selection in a nonlinear inverted-U-shaped way depending on the inputs and the network state (*Figure 7*). Indeed, experimental results have suggested that the amplitude of activities of D2 pathway is pivotal to the behavioral outcome (*Meng et al., 2018*), consistent with our proposed 'nonlinear' control of D2 pathway over action selection. These results of various functional assemblies defied previous basal ganglia models in which either direct or indirect pathway has been treated as one uniform population and assigned with a single function in controlling action.

In the 'Triple-control' model, we posited the collateral connections among striatal D2-SPNs and its short-term plasticity could serve as an operational mechanism for the dominant module switching. However, besides these well-known striatal local connections as one of the simplest possible mechanisms, other anatomical circuits within basal ganglia circuitry could potentially fulfill this functional role alone or additionally as well. For example, striatal D2-SPNs project to external global pallidus (GPe) through striatopallidal pathway, and meanwhile they receive arkypallidal projections from GPe to both the striatal SPNs and interneurons (*Abdi et al., 2015*; *Fujiyama et al., 2016*; *Mallet et al., 2016*). It is thus also possible that the dynamic interaction between 'Co-activation' and 'Go/No-go' modules is mediated through di-synaptic or tri-synaptic modulation with GPe and/or striatal interneurons. Furthermore, in theory this dynamic interaction between 'Co-activation' and 'Go/No-go' modules can also occur outside striatum in the downstream nuclei including GPe and SNr, given their specific neuronal subpopulations receiving inputs from corresponding striatal D2-SPNs subgroups and proper collateral connections within the nuclei (*Atherton et al., 2013*; *Cazorla et al., 2014*; *Fujiyama et al., 2011*; *Lee et al., 2020*; *Wu et al., 2000*). Considering the crucial role of dopamine in basal ganglia circuitry, the new 'Triple-control' model can also reproduce our previous experiments results on the effect of nigrostriatal dopamine on action selection (*Howard et al., 2017*). Importantly, it unveils that there is an inverted-U-shaped relationship between dopamine concentration change and action selection (*Cools and D'Esposito, 2011*). The model simulation suggests that

while moderate dopamine increase improves decision making, too much dopamine changes, either increase or decrease, dramatically impair the choice behavior. These results might be able to explain some of behavioral observations involving in obscure decision making under the influence of addictive substances.

Our findings also have important implications in many neurological and psychiatric diseases. It was known that the loss of dopamine leads to hyperactivity of D2-SPNs and disruption of local D2-SPNs collaterals in Parkinson's disease (*Taverna et al., 2008*; *Wei and Wang, 2016*). These alterations will not only break the balance of direct vs. indirect pathway, but also disrupt the multiple dynamic controls from the indirect pathway. The action selection will thus be largely problematic, even with L-DOPA treatment, which might restore the dopamine partially but not necessarily the altered basal ganglia circuitry and its circuit dynamics (*Bastide et al., 2015*). The current 'Triple-control' model also provides some mechanistic insights into the inhibitory control deficits observed schizophrenia (*Taverna et al., 2008*; *Wei and Wang, 2016*). For instance, an increase in the density and occupancy of the striatal D2 receptors (D2R) has been frequently reported in schizophrenia patients (*Abi-Dargham et al., 2000*; *Howes and Kapur, 2009*; *Laruelle et al., 1997*; *Wong et al., 1986*). Many antipsychotic medications primarily aim to block the D2R (*la Fougère et al., 2005*; *Lally and MacCabe, 2015*; *Yokoi et al., 2002*), but the drug dose is the key to the treatment and severer adverse effects are associated with overdose of D2R antagonism (*Levine and Ruha, 2012*). In addition, prolonged exposure to antipsychotics often causes extrapyramidal symptoms, including Parkinsonian symptoms and tardive dyskinesia (*Jarskog et al., 2007*; *Seeman, 2002*). The dose-dependent effects when modulating D2R were also found in cognitive functions such as serial discrimination, in which relatively low and high dose of D2R agonist in striatum impairs the performance in the discrimination task, while the intermediate dose of D2R agonist produces significant improvement (*Cools and D'Esposito, 2011*; *Goldman-Rakic et al., 2000*; *Horst et al., 2019*; *Mattay et al., 2003*). These observations thus further underscore the dynamic interplays and complexity of basal ganglia pathways in action control, as demonstrated in current study and the new 'Triple-control' functional model.

## Methods

### Key resources table

| Reagent type (species) or resource | Designation | Source or reference | Identifiers | Additional information |
|---|---|---|---|---|
| Strain, strain background (adeno-associated virus) | AAV9-FLEX-DTR-GFP | Salk GT3 Core | N/A | |
| Strain, strain background (adeno-associated virus) | AAV5-EF1a-DIO-hChR2(H134R)-mCherry | University of North Carolina Vector Core | N/A | |
| Strain, strain background (adeno-associated virus) | AAV9-EF1a-DIO-hChR2(H134R)-eYFP | University of Pennsylvania Vector Core | Cat# AV-9-20298P | |
| Strain, strain background (adeno-associated virus) | AAV5-EF1a-DIO-eNpHR3.0-eYFP | University of North Carolina Vector Core | N/A | |
| Chemical compound, drug | Muscimol, GABAA receptor agonist | Sigma-Aldrich | Cat# M1523 | |
| Chemical compound, drug | Diphtheria toxin | List Biological Labs | Part# 150 | |
| Strain, strain background (*Mus musculus*) | Mouse: C57BL/6 | Envigo/Harlan | Code: 044 | |
| Strain, strain background (*Mus musculus*) | Mouse: NR1f/f (B6.129S4-Grin1tm2Stl/J) | Jackson Laboratory | Stock# 005246 | |
| Strain, strain background (*Mus musculus*) | Mouse: Ai32 (B6;129S-Gt(ROSA)26Sortm32(CAG-COP4*H134R/EYFP)Hze/J) | Jackson Laboratory | Stock# 012569 | |

*Continued on next page*

*Continued*

| Reagent type (species) or resource | Designation | Source or reference | Identifiers | Additional information |
|---|---|---|---|---|
| Strain, strain background (*Mus musculus*) | Mouse: D1-cre (B6.FVB(Cg)-Tg(Drd1a-cre)EY217Gsat/Mmucd) | MMRRC | RRID: MMRRC_034258-UCD | |
| Strain, strain background (*Mus musculus*) | Mouse: A2a-cre (B6.FVB(Cg)-Tg(Adora2a-cre)KG139Gsat/Mmucd) | MMRRC | RRID: MMRRC_036158-UCD | |
| Software, algorithm | GraphPad Prism | GraphPad Software | Version 7.03 | https://www.graphpad.com/scientific-software/prism/ |
| Software, algorithm | MATLAB | MathWorks | R2013a | https://www.mathworks.com/products/matlab.html |
| Software, algorithm | Med-PC | Med Associates | Cat# SOF-735 | https://med-associates.com/product/med-pc-v/ |
| Software, algorithm | Offline Sorter | Plexon | Version 3.3.3 | https://plexon.com/products/offline-sorter/ |
| Software, algorithm | OmniPlex | Plexon | Version 1.4.5 | https://plexon.com/products/omniplex-software/ |
| Software, algorithm | EthoVision | Noldus | Version 8.5 | |
| Other | Med Associates operant chamber | Med Associates | Cat# MED-307W-D1 | |
| Other | Electrode Array | Innovative Neurophysiology | N/A | |
| Other | 473 nm laser | LaserGlow Technologies | N/A | |
| Other | 532 nm laser | LaserGlow Technologies | N/A | |

## Animals

All experiments were approved by the Salk Institute Animal Care, and done in accordance with NIH guidelines for the Care and Use of Laboratory Animals. Experiments were performed on both male and female mice, at least 2 months of age, housed on a 12 hr light/dark cycle. C57BL/6J mice were purchased from the Jackson Laboratory at 8 weeks of age and used as wild-type mice. BAC transgenic mice expressing cre recombinase under the control of the dopamine D1 receptor (referred as D1-cre, GENSAT: EY217; minimal labeling in cortex; mostly dorsal labeling in striatum) or the A2a receptor (referred to as A2a-cre, GENSAT: KG139) promoter were obtained from MMRRC, and either crossed to C57BL/6 or Ai32 (012569) mice obtained from Jackson Laboratory (*Cui et al., 2013*; *Geddes et al., 2018*; *Jin et al., 2014*; *Madisen et al., 2012*; *Tecuapetla et al., 2016*). Striatal neuron-type-specific NMDAR1-knockout (referred to as NR1-KO) and littermate controls were generated by crossing D1-cre and A2a-cre mice with NMDAR1-loxP (also denoted as Grin1 flox/flox in the Jackson Laboratory database) mice. The behavioral experiments using NR1-KO mice were performed on 8- to 12-week-old D1/A2a-cre + / NMDAR1-loxP homozygous mice and their littermate controls, including D1/A2a-cre +, D1/A2a-cre + / NMDAR1-loxP heterozygous and NMDAR1-loxP homozygous mice. There was no difference between the three control groups, so the data were combined.

## Behavior task

Mice were trained on a temporal bisection task in the operant chamber (21.6 cm L × 17.8 cm W × 12.7 cm H), which was isolated within a sound attenuating box (Med-Associates, St. Albans, VT, USA). The food magazine was located in the middle of one wall, and two retractable levers were located to the left and right side of the magazine. A house light was (3 W, 24 V) mounted on the opposite wall of the magazine. Sucrose solution (10%) was delivered into a metal bowl in the magazine through a syringe pump. When a training session started, the house light was turned on and two levers were extended. After a random time interval (30 s on average), left and right levers were retracted and extended simultaneously. Mice were able to make a choice by pressing either left or right lever. Only the very first lever press after levers extension was registered as animals' choice. If the interval between the levers retraction and extension was 2 s, then only the left lever press was active to trigger the sucrose reward; if the interval between the lever retraction and extension was 8 s, then only the right lever

press was active to trigger the sucrose reward (*Howard et al., 2017*). There was no punishment when mice made an unrewarded choice. Two s-trial and 8 s-trial were randomized and interleaved by random ITI (30 s on average). The mice were also trained in the reversed 2–8 s task. In the reversed 2–8 s task, if the interval between the levers retraction and extension was 2 s, then only the right lever press was active to trigger the sucrose reward; if the interval between the lever retraction and extension was 8 s, then only the left lever press was active to trigger the sucrose reward. Representative behavioral tracks were captured by EthoVision (Noldus).

## Behavior training

Mice were placed on food restriction throughout the training, and fed daily after the training sessions with ~2.5 g of regular food to allow them to maintain a body weight of around 85% of their baseline weight. Training started with continuous reinforcement (CRF), in which animals obtained a reinforcer after each lever press. The session began with the illumination of the house light and extension of either left or right lever, and ended with the retraction of the lever and the offset of the house light. On the first day of CRF, mice received 5 reinforcers on left and right lever. On the second day of CRF, mice received 10 reinforcers on left and right lever. On the third day of CRF, mice received 15 reinforcers on left and right lever. The order of left lever CRF and right lever CRF on each day was randomized across all the CRF training days. After the training of CRF, animals started on the temporal bisection task (day 1). Mice were trained in the temporal bisection task for 14 consecutive days. On each day, there were 240 trials with 2 s-trial and 8 s-trial randomly intermixed at 50:50. After 14 days training, mice received an interval discrimination test, in which 20% of 2 s/8 s trials were replaced by probe trials. In probe trials, the levers retraction intervals were randomly selected from 2.3 s, 3.2 s, 4 s, 5 s, and 6.3 s. Neither choice in the probe trials was rewarded. Mice received 4 days of test, interleaved by training days without probes. The animals were trained daily without interruption and every day the training started approximately at same time (*Howard et al., 2017*). All timestamps of lever presses, magazine entries, and licks for each animal were recorded with 10 ms resolution. The training chambers and procedures for NR1-KO mice and their littermate controls were exactly the same for C57BL/6J mice.

For the reversed task training, mice were trained in both the 2–8 s control task and reversed version of 2–8 s task on the same day for at least 14 days. During each day, mice were trained in the 2–8 s task first, and then mice were put back in the home cage for a 3–4 hr rest. After the rest period, the same mice were trained in the reversed 2–8 s task. The order of these two tasks is fixed throughout the 14 days' training.

## Surgery

For in vivo electrophysiological data recording, each mouse was chronically implanted with an electrode array which consists of an array of 2 rows × 8 columns platinum-coated tungsten microwire electrodes (35 µm diameter) with 150 µm spacing between microwires in a row, and 250 µm spacing between 2 rows. The craniotomies were made at the following coordinates: 0.5 mm rostral to bregma and 1.5 mm laterally for dorsal striatum; 3.4 mm caudal to bregma and 1.0 mm laterally for SNr (*Jin and Costa, 2010*; *Jin et al., 2014*). During surgeries, the electrode arrays were gently lowered ~2.2 mm from the surface of the brain for dorsal striatum and ~4.3 mm for SNr, while simultaneously monitoring neural activity. Final placement of the electrodes was monitored online during the surgery based on the neural activity, and then confirmed histologically at the end of the experiment after perfusion with 10% formalin, brain fixation in a solution of 30% sucrose and 10% formalin, followed by cryostat sectioning (coronal slices of 40–60 µm). For striatum recording, we implanted 11 mice in the left hemisphere and 8 mice in the right hemisphere. For the SNr recording, we implanted 5 mice in the left hemisphere and 4 mice in the right hemisphere.

For the cell-type identification in striatum, the cre-inducible adeno-associated virus (AAV) vector carrying the gene encoding the light-activated cation channel ChR2 and a fluorescent reporter (DIO-ChR2-YFP/DIO-ChR2-mCherry) was stereotactic injected into the dorsal striatum of D1-Cre or A2a-Cre mice, enabling cell-type-specific expression of ChR2 in striatal D1-expressing or D2-expressing projection neurons (at exactly the same coordinates of electrode array implantation in striatum stated above). DIO-ChR2-YFP/DIO-ChR2-mCherry virus (1 µl, one site, $10^{12}$ titer) was injected through a micro-injection Hamilton syringe, with the whole injection taking ~10 min in total. The syringe needle was left in the position for 5–10 min after the injection and then slowly moved out. Following viral

injections or for mice genetically expressing ChR2 under cre control (D1-Ai32, A2a-Ai32), electrode was implanted as previously described (*Geddes et al., 2018*; *Jin et al., 2014*). The electrode array was the same as used for dorsal striatum recording, but with a guiding cannula attached (Innovative Neurophysiology) terminating ~300 μm above the electrode tips, and was implanted into the same site after virus injection, allowing for simultaneous electrophysiological recording and light stimulation. Following the implantation, a medal needle was inserted in the cannula and mice were placed in the home cage for 2 weeks, allowing both viral expression and surgery recovery, before further training and recording experiments.

For the optogenetic manipulation in striatum, we injected the AAV carrying the gene for coding ChR2 (DIO-ChR2-YFP/DIO-ChR2-mCherry) or Halorhodopsin (DIO-eNpHR3.0-eYFP). Virus was injected bilaterally at 0.5 mm rostral to bregma, 2 mm laterally and ~2.2 mm from the surface of the brain with 1 μl per site. 10 min after the virus injection, we bilaterally implanted optical fiber units in dorsal striatum to the same site as virus injection. An optical fiber unit was made by threading a 200 μm diameter, 0.37 NA optical fiber (Thor Labs) with epoxy resin into a plastic ferrule (*Geddes et al., 2018*; *Howard et al., 2017*). Optical fiber units were then cut and polished before the implantation.

For muscimol infusion in striatum, we bilaterally implanted cannulas (Plastics One, VA, USA) in wild-type mice to the site at 0.5 mm rostral to bregma, 2 mm laterally and ~2.2 mm from the surface of the brain. After the implantation, cannulas were covered by dummy cannulas. Mice were placed in the home cage for 2 weeks, allowing surgery recovery, before further training and muscimol experiments.

For striatal neuron-type-specific ablation experiments, D1-cre and A2a-cre mice were stereotaxically injected with a cre-inducible AAV carrying the DTR (*Azim et al., 2014*; *Geddes et al., 2018*) (AAV9-FLEX-DTR-GFP; Salk GT3 Core, CA, USA). Virus was injected in eight different sites. We used two sets of AP/ML coordinates for each hemisphere followed by two DV depths at each AP/ML site. The coordinates were +0.9 mm AP, ±1.6 mm ML, –2.2 and –3.0 mm DV and 0.0 mm AP, ±2.1 mm ML, –2.2 and –3.0 mm DV. A Hamilton syringe was used to inject 1 μl at the four –3.0 mm DV sites and another 0.5 μl at the four –2.2 mm DV sites for a total of 3 μl injected per hemisphere. Following each injection, the needle was left in place for ~5 min and then raised over ~5 min. This same protocol was used for each injection site.

## Muscimol infusion

We daily trained wild-type mice with guide cannulas (Plastics One, VA, USA) implanted until they achieved at least 80% correct rate for 3 consecutive days, we started muscimol infusion experiments. Muscimol was dissolved in saline before infusion (Sigma-Aldrich; 0.05 μg/μl). For the infusions, mice were briefly anesthetized with isoflurane and injection cannulas (Plastics One, VA, USA) were bilaterally inserted into the guide cannulas, with the injection cannulas projecting 0.1 mm beyond the implanted guide cannulas. Each injection cannula was attached to an infusion pump (BASi, IN, USA) via polyethylene tubing. Animals were bilaterally infused with 200 nl of liquid (saline or muscimol) followed by a 5-min waiting period before removal of the infusion cannulas. Mice were returned to their home cage and started in the behavioral task 30 min after infusion (*Geddes et al., 2018*). To estimate the effects of muscimol on choice, we repeated saline controls and muscimol infusions at least three times on a single mouse to gain enough probe trials for psychometric curve fitting.

## DTR-mediated cell ablation

For striatal neuron-type-specific ablation experiments, D1-cre and A2a-cre were injected with AAV9-FLEX-DTR-GFP in striatum using the same coordinates described above. After 3-week recovery, mice were food-restricted and, following completion of CRF, underwent training in the 2–8 s task for 2 weeks. Immediately after day 14 of 2–8 s task training, mice were randomly divided into control and treatment groups. Treatment mice were administered mice 1 μg of DT dissolved in 300 μl of phosphate buffered saline (PBS) via intraperitoneal (IP) injection on 2 consecutive days (*Azim et al., 2014*; *Geddes et al., 2018*), whereas control mice received IP injections of PBS. To allow for neuronal ablation, animals were stopped in behavioral training and placed back on food. Animals resumed 2–8 s task training with probe trials 14 days after the first DT or PBS injection.

## Neural recordings during the task

The mice with electrode array implanted were trained with exactly the same procedure as described above. When mice performed the 2–8 s task with 80% correct rate for 3 consecutive days, we connected the array with recording cable and continued training until mice adapted to the mechanics of the recording cable and were able to maintain the correct rate greater than 80% (*Howard et al., 2017*).

Neural activity was recorded using the MAP system (Plexon Inc, TX, USA). The spike activities were initially online sorted using a sorting algorithm (Plexon Inc, TX, USA). Only spikes with a clearly identified waveforms and relatively high signal-to-noise ratio were used for further analysis. After the recording session, the spike activities were further sorted to isolate single units by offline sorting software (Plexon Inc, TX, USA). Single units displayed a clear refractory period in the inter-spike interval histogram, with no spikes during the refractory period (larger than 1.3 ms) (*Geddes et al., 2018*; *Howard et al., 2017*; *Jin and Costa, 2010*; *Jin et al., 2014*). All the timestamps of animal's behavioral events were recorded as TTL pulses which were generated by a Med-Associates interface board and sent to the MAP recording system through an A/D board (Texas Instrument Inc, TX, USA). The animal's behavioral timestamps during the training session were synchronized and recorded together with the neural activity.

## Neural dynamic analysis

The animal's behavior taking place during the lever retraction time period was critical to the choice to be made, so we focused on the analysis of the neural activity from levers retraction to levers extension. Neuronal firings aligned to lever retraction were averaged across trials in 20 ms bins, and smoothed by a Gaussian filter (Gaussian filter window size = 10, standard deviation = 5) to construct the peristimulus time histogram (PETH). The neurons showing significant firing changes during the lever retraction period were defined as task-related neurons (ANOVA); those showing no significant changes were defined as non-task-related neurons, which were not included in the further dynamic analysis.

During 2 s trials, mice behaved exactly the same as they did during the 0–2 s period in the rewarded 8 s trials, so we mainly analyzed firing activities in 8 s trials. To avoid confounding effect by the sensory responses triggered by the lever retraction, only neural activity from 1 s to 8 s following lever retraction were included (*Howard et al., 2017*). Then, we calculated FRI based on the PETH from 1 s to 8 s for each individual neuron as follows:

$$\text{FRI} = \frac{\text{PETH} - \text{mean}\left(\text{PETH}\right)}{\text{std}\left(\text{PETH}\right)}$$

We then used principal component analysis and classification algorithm, a build-in toolbox in Matlab, to classify the task-related neurons based on types of dynamics. For striatum and SNr, we used the same algorithm to classify neurons, and we found the same types of dynamics in striatum and SNr: Type 1, monotonic decreasing; Type 2, monotonic increasing; Type 3, peak at around 4 s; Type 4, trough at around 4 s.

## Cell-type classification

In dorsal striatum, we classified neurons as putative SPNs if they showed waveform trough half-width between 100 µs and 250 µs and the baseline firing rate less than 10 Hz. In SNr, neurons with firing rate higher than 15 Hz were classified as putative SNr GABA neurons, which are most likely the SNr projection neurons, because the percentage of GABAergic interneurons in the SNr is rather small (*Deniau et al., 2007*; *Jin and Costa, 2010*).

To further identify the D1 and D2 SPNs in striatum, we utilized cre-loxp technique to exclusively express ChR2 on D1-SPNs or D2-SPNs by injecting the AAV-DIO-ChR2-YFP/AAV-DIO-ChR2-mCherry virus into dorsal striatum or genetically express ChR2 by D1-Ai32 and A2a-Ai32. Optical stimulation on ChR2-expressed cells is able to directly evoke spiking activity with short latency (*Geddes et al., 2018*; *Jin and Costa, 2010*; *Jin et al., 2014*). Before the training session, we connected the recording cable to the electrode array for neuronal recording and inserted an optic fiber through the cannula attached to the array to conduct light into striatum for light stimulation. For better monitoring of the same cells stably during behavioral training and the later optogenetic identification, the optic fiber

was well fixed to the array. After each training session, we delivered blue light stimulation through the optic fiber from a 473 nm laser (Laserglow Technologies) via a fiber-optic patch cord, and simultaneously recorded the neuronal responses, to testify the molecular identity of cells previously recorded during the behavioral training. The stimulation pattern was 100 ms pulse width with 4 s interval. The stimulation pattern was repeatedly delivered for 100 trials. We very carefully regulated the laser power to a relatively low level for each individual recording session which was strong enough to evoke reliable spikes in a small population of neurons recorded from certain electrodes, since high laser powers usually induced an electrical signal much larger and very different from the spike waveforms previously recorded in the same electrode, presumably resulting from synchronized activation of a large population of cells surrounding the electrode. For neuron identification in different sessions in the same mouse, substantial effort was made to optimize the position of optic fiber to identify those units recorded from different electrodes and that were not being able to be identified in the previous session. The final laser power used for reliable identification of D1/D2-SPNs was between 1.0 mW and 1.5 mW measured at the tip of the optical fiber (slightly varying for different mice and different sessions). Only those units showing very short (≤6 ms) response latency to light stimulation and exhibiting exactly the same spike waveforms ($R≥0.95$, Pearson's correlation coefficient) during the behavioral performance and light response were considered as direct light-activated and cre recombinase positive neurons thus D1-SPNs or D2-SPNs (*Geddes et al., 2018*; *Howard et al., 2017*; *Jin and Costa, 2010*). Strict criteria were employed to minimize the possibility of false positives (with the risk of increasing false negatives, and hence having to perform more recordings/mice to achieve the same number of neurons).

## Optical stimulation during the task

For optogenetic manipulation experiments, mice were injected with AAV carrying were pre-trained in 2–8 s task for 2 weeks and bilaterally implanted with optic fibers. After achieving a correct rate of 80%, stimulation trials began. D1-SPNs and D2-SPNs neurons were stimulated or inhibited bilaterally in 50% of trials using a single pulse of light (Laserglow, 473 nm, 5 mW, 1 s constant for ChR2 experiments; Laserglow, 532 nm 10 mW, 1 s constant for Halorhodopsin experiments). Rewards were delivered only at correct responses during 2 s and 8 s trials. Within 50% of any type of trials, mice were optogenetically stimulated (or inhibited) for 1 s before lever extension (*Howard et al., 2017*). Mice only received stimulation (or inhibition) once per trial. Sessions with correct rate below 75% for control trials were excluded from further analysis.

## Computational model

We constructed a neuronal network model, including cortico-basal ganglia circuitry, to simulate the behavioral effects of ablation and optogenetic manipulation on SPNs. Specifically, cortical information corresponding to left or right choice is sent to D1- and D2-SPNs associated with these two action options (*Lo and Wang, 2006*; *Wang, 2002*). One-way collateral inhibition is added between D2 SPNs subgroups. Signals from D1- and D2-SPNs eventually converge to two separate SNr populations through distinct pathways (*Hikosaka et al., 2000*; *Jin et al., 2014*; *Mink, 2003*), and exert opposing effects on SNr activity (*Smith et al., 1998*). Behavioral output is then determined by the dominant activity between the mutually inhibiting left and right SNr populations (*Mailly et al., 2003*), which could control the final motor output either through brainstem circuits or motor cortices (*Aoki et al., 2019*; *Hikosaka, 2007*; *Lo and Wang, 2006*; *Redgrave et al., 1999*). Here for simplicity, other basal ganglia nuclei such as globus pallidus and subthalamic nucleus are not included in the model.

Cortical neurons firing activities are defined as:

$$f_{left}^{cortex}(t) = k_{left}^{cortex} e^{-t_m \cdot t} + I_{noise}^{left}(t)$$

$$f_{right}^{cortex}(t) = k_{right}^{cortex} e^{-t_m \cdot t} + I_{noise}^{right}(t)$$

where $k_{left}^{cortex} = 2$, $k_{right}^{cortex} = -2$, $t_m = 0.4$ and $I_{noise}(t)$ is defined as Gaussian white noise (mean($I_{noise}^{left}$)=1, mean($I_{noise}^{right}$)=2, SD =0.01).

Dopamine neuron firing activities is defined as:

$$f_{DA}(t) = k_{DA} e^{-t_{DA} \cdot t} + I_{noise}^{DA}(t)$$

where $k_{DA} = 3$, $t_{DA} = 0.4$, and $I_{noise}^{DA}(t)$ is defined as Gaussian white noise (mean ($I_{noise}^{DA}$)=1, SD =0.01).

Neuronal activities of D1-SPNs are defined as:

$$\tau \frac{df_{left}^{D1}(t)}{dt} = w_0 \left(E - f_{left}^{D1}(t)\right) + \tilde{w}_{D1\_left} f_{left}^{cortex}(t) + w_{D1} f_{DA}(t) + I_{noise}(t)$$

$$\tau \frac{df_{right}^{D1}(t)}{dt} = w_0 \left(E - f_{right}^{D1}(t)\right) + \tilde{w}_{D1\_right} f_{right}^{cortex}(t) + w_{D1} f_{DA}(t) + I_{noise}(t)$$

where $w_0 = 1$, $E = 20$, $\tilde{w}_{D1\_left} = 3$, $\tilde{w}_{D1\_right} = 6$, $w_{D1} = 2$, $\tau = 0.1$, $I_{noise}(t)$ is defined as Gaussian white noise (mean ($I_{noise}(t)$)=0, SD =0.5).

Neuronal activities of D2-SPNs in 'Co-activation' module (labeled as D2-SPN 1) are defined as:

$$\tau \frac{df_{left1}^{D2}(t)}{dt} = w_0 \left(E - f_{left1}^{D2}(t)\right) + \tilde{w}_{D2\_left} f_{left}^{cortex}(t) + w_{D2} f_{DA}(t) + I_{noise}(t)$$

$$\tau \frac{df_{right1}^{D2}(t)}{dt} = w_0 \left(E - f_{right1}^{D2}(t)\right) + \tilde{w}_{D2\_right} f_{right}^{cortex}(t) + w_{D2} f_{DA}(t) + I_{noise}(t)$$

where $w_0 = 1$, $E = 21$, $\tilde{w}_{D2\_left} = 5$, $\tilde{w}_{D2\_right} = 5$, $w_{D2} = -0.3$, $\tau = 0.1$, $I_{noise}(t)$ is defined as Gaussian white noise (mean ($I_{noise}(t)$)=0, SD =0.5).

Neuronal activities of D2-SPNs in 'Go/No-go' module (labeled as D2-SPN 2) are defined as:

$$\tau \frac{df_{left2}^{D2}(t)}{dt} = w_0 \left(E - f_{left2}^{D2}(t)\right) + \tilde{w}_{D2\_left} f_{left}^{cortex}(t) + w_{D2\_left} S_{left}(t) f_{right1}^{D2}(t) + w_{D2} f_{DA}(t) + I_{noise}(t)$$

$$\tau \frac{df_{right2}^{D2}(t)}{dt} = w_0 \left(E - f_{right2}^{D2}(t)\right) + \tilde{w}_{D2\_right} f_{right}^{cortex}(t) + w_{D2\_right} S_{right}(t) f_{left1}^{D2}(t) + w_{D2} f_{DA}(t) + I_{noise}(t)$$

where $w_0 = 1$, $E = 21$, $\tilde{w}_{D2\_left} = 5$, $\tilde{w}_{D2\_right} = 5$, $w_{D2\_left} = -0.7$, $w_{D2\_right} = -0.5$, $w_{D2} = -0.3$, $\tau = 0.1$, $I_{noise}(t)$ is defined as Gaussian white noise (mean ($I_{noise}(t)$)=0, SD =0.5). $S_{left}(t)$ and $S_{right}(t)$ are short-term depression functions:

$$S_{left}(t) = 3 / \left(1 + e^{0.3\left(f_{right1}^{D2}(t) - 15\right)}\right)$$

$$S_{right}(t) = 3 / \left(1 + e^{0.3\left(f_{left1}^{D2}(t) - 15\right)}\right)$$

SNr neurons receive striatal inputs as well as the local inhibitory inputs from other SNr neurons. The SNr activities are defined as:

$$\tau_{SNr} \frac{df_{left}^{SNr}(t)}{dt} = w_0 \left(E - f_{left}^{SNr}(t)\right) + \tilde{w}1_{SNr\_left} f_{left}^{D1}(t) + \tilde{w}2_{SNr} f_{right1}^{D2}(t) + \tilde{w}3_{SNr} f_{left2}^{D2}(t) + w_{left}^{SNr} f_{right}^{SNr}(t) + I_{noise}^{s}(t)$$

$$\tau_{SNr} \frac{df_{right}^{SNr}(t)}{dt} = w_0 \left(E - f_{right}^{SNr}(t)\right) + \tilde{w}1_{SNr\_right} f_{right}^{D1}(t) + \tilde{w}2_{SNr} f_{left1}^{D2}(t) + \tilde{w}3_{SNr} f_{right2}^{D2}(t) + w_{right}^{SNr} f_{left}^{SNr}(t) + I_{noise}^{s}(t)$$

where $w_0 = 1$, $E = 40$, $\tilde{w}1_{SNr\_left} = -0.1$, $\tilde{w}1_{SNr\_right} = -0.105$, $\tilde{w}2_{SNr} = 0.15$, $\tilde{w}3_{SNr} = 0.07$, $w_{left}^{SNr} = -0.027$, $w_{right}^{SNr} = -0.01$, $\tau_{SNr} = 0.2$. $I_{noise}^{s}(t)$ is defined as Gaussian white noise (mean ($I_{noise}^{s}$)=0, SD = 0.3).

The time-dependent choice $C(t)$ is then determined by SNr outputs $f_{left}^{SNr}(t)$ and $f_{right}^{SNr}(t)$ as follows:

$$C(t) = \begin{cases} left\ choice\ (short-duration\ choice), & f_{left}^{SNr}(t) - f_{right}^{SNr}(t) < 0 \\ right\ choice\ (long-duration\ choice), & f_{left}^{SNr}(t) - f_{right}^{SNr}(t) \geq 0 \end{cases}$$

For optogenetic manipulation of striatal neurons, the stimulation pattern is defined as:

$$F_{activation}(t) = \begin{cases} amp, & t_s \leq t \leq t_s + 1 \\ 0, & t < t_s \text{ or } t > t_s + 1 \end{cases}$$

and for inhibition, the pattern is defined as:

$$F_{inhibition}(t) = \begin{cases} -amp, & t_s \leq t \leq t_s + 1 \\ 0, & t < t_s \text{ or } t > t_s + 1 \end{cases}$$

where $t_s$ is the onset of stimulation/inhibition, which lasts for 1 s. *amp* is defined as the strength of the optogenetic manipulation within the range of [1, 25].

To add D1-D1 collateral connections to the 'Triple-control' model, the neuronal activities of D1-SPNs are defined as:

$$\tau \frac{df_{left}^{D1}(t)}{dt} = w_0\left(E - f_{left}^{D1}(t)\right) + \widetilde{w}_{D1_{left}} f_{left}^{cortex}(t) + w_{D1\_right} f_{right}^{D1}(t) + w_{D1} f_{DA}(t) + I_{noise}(t)$$

$$\tau \frac{df_{right}^{D1}(t)}{dt} = w_0\left(E - f_{right}^{D1}(t)\right) + \widetilde{w}_{D1_{right}} f_{right}^{cortex}(t) + w_{D1\_left} f_{left}^{D1}(t) + w_{D1} f_{DA}(t) + I_{noise}(t)$$

where $w_{D1\_Left} = -0.3, w_{D1\_right} = -0.3$.

To add D1-D2 collateral connections to the 'Triple-control' model, the neuronal activities of D2-SPNs in 'Go/No-go' module (labeled as D2-SPN 2) are defined as:

$$\tau \frac{df_{left2}^{D2}(t)}{dt} = w_0\left(E - f_{left2}^{D2}(t)\right) + \widetilde{w}_{D2\_left} f_{left}^{cortex}(t) + w_{D2_{left}} S_{left}(t) f_{right1}^{D2}(t) + w_{D1_{left}} f_{left}^{D1}(t) + w_{D2} f_{DA}(t) + I_{noise}(t)$$

$$\tau \frac{df_{right2}^{D2}(t)}{dt} = w_0\left(E - f_{right2}^{D2}(t)\right) + \widetilde{w}_{D2\_right} f_{right}^{cortex}(t) + w_{D2\_right} S_{right}(t) f_{left1}^{D2}(t) + w_{D1\_right} f_{right}^{D1}(t) + w_{D2} f_{DA}(t) + I_{noise}(t)$$

where $w_{D1\_left} = -0.3, w_{D1\_right} = -0.3$.

To add D2-D1 collateral connections to the 'Triple-control' model, the neuronal activities of D1-SPNs are defined as:

$$\tau \frac{df_{left}^{D1}(t)}{dt} = w_0\left(E - f_{left}^{D1}(t)\right) + \widetilde{w}_{D1_{left}} f_{left}^{cortex}(t) + w_{D2\_left} f_{left2}^{D2}(t) + w_{D1} f_{DA}(t) + I_{noise}(t)$$

$$\tau \frac{df_{right}^{D1}(t)}{dt} = w_0\left(E - f_{right}^{D1}(t)\right) + \widetilde{w}_{D1_{right}} f_{right}^{cortex}(t) + w_{D2\_right} f_{right2}^{D2}(t) + w_{D1} f_{DA}(t) + I_{noise}(t)$$

where $w_{D2\_left} = -0.3, w_{D2\_right} = -0.3$.

All the modeling programs were coded in Matlab.

## Psychometric curve fitting

Psychometric curves for behavioral data and for theoretical curves were fit using the following equation (*Brunton et al., 2013*; *Howard et al., 2017*):

$$y = a + \frac{b}{1 + e^{\frac{c-x}{d}}}$$

where *a* is the percentage of long-lever selection during short duration trials, *b* is the difference between *a* and the percentage of long-lever selection during long duration trials, *c* is the x-intercept where long-duration selection equals 0.5, and *d* is the rate of increase or decrease in the curve (slope). These can be interpreted as change in overall choice, long-duration choice, time, and sensitivity, respectively (*Brunton et al., 2013*).

## Statistical procedures

Statistics for the wild-type and KO mice learning data were performed on the basis of values for each mouse per day. One-way and two-way repeated-measures ANOVA were used to investigate general main effects; and paired or unpaired t-tests were used in all planned and post hoc comparisons. Z-test was used for the comparison of neuron proportions (*Sheskin, 2003*). Statistics for the optogenetic data were performed on the basis of control and stimulated values for each mouse per stimulation condition. Statistical analyses were conducted in Matlab using the statistics toolbox (The MathWorks Inc, MA USA) and GraphPad Prism 7 (GraphPad Software Inc, CA, USA). Results are presented as mean ± SEM for behavior readouts and the neuronal recording data. $p < 0.05$ was considered significant. All statistical details are located within the figure legends. The number of animals used in each experiment and the number of neurons are specified in the text and figure legend.

## Acknowledgements

The authors would like to thank Tom Jessell and members of Jin lab for discussion and comments on the manuscript. This research was supported by grants from the NIH (R01NS083815), the Dystonia Medical Research Foundation, and the McKnight Memory and Cognitive Disorders Award to XJ.

## Additional information

### Funding

| Funder | Grant reference number | Author |
| --- | --- | --- |
| National Institutes of Health | R01NS083815 | Xin Jin |
| Dystonia Medical Research Foundation | | Xin Jin |
| McKnight Endowment Fund for Neuroscience | McKnight Memory and Cognitive Disorders Award | Xin Jin |

The funders had no role in study design, data collection and interpretation, or the decision to submit the work for publication.

### Author contributions

Hao Li, Conceptualization, Data curation, Formal analysis, Investigation, Visualization, Methodology, Writing - original draft, Writing - review and editing; Xin Jin, Conceptualization, Resources, Supervision, Funding acquisition, Writing - original draft, Project administration, Writing - review and editing

### Author ORCIDs

Hao Li ⓘ http://orcid.org/0000-0002-6312-4276
Xin Jin ⓘ http://orcid.org/0000-0002-1106-4013

### Ethics

All the experiments were conducted at the Salk Institute for Biological Studies according to NIH guidelines, and all the protocols were approved by their Institutional Animal Care and Use Committee (12-00032).

Reviewer #1 (Public Review): https://doi.org/10.7554/eLife.87644.3.sa1
Reviewer #2 (Public Review): https://doi.org/10.7554/eLife.87644.3.sa2
Author Response https://doi.org/10.7554/eLife.87644.3.sa3

## Additional files

### Supplementary files
• MDAR checklist

## Data availability

Data and code are available at Dryad: https://doi.org/10.5061/dryad.gxd2547s1.

The following dataset was generated:

| Author(s) | Year | Dataset title | Dataset URL | Database and Identifier |
|---|---|---|---|---|
| Li H, Jin X | 2023 | Experimental data | https://doi.org/10.5061/dryad.gxd2547s1 | Dryad Digital Repository, 10.5061/dryad.gxd2547s1 |

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
