## [Editor Report · eLife assessment]

In this **valuable** manuscript Li & Jin record from the substantial nigra and dorsal striatum to identify subpopulations of neurons with activity that reflects different dynamics during action selection, and then use optogenetics in transgenic mice to selectively inhibit or excite D1- and D2- expressing spiny projection neurons in the striatum, demonstrating a causal role for each in action selection in an opposing manner. They provide **solid** evidence for the argument that their findings cannot be explained by current models and propose a new 'triple control' model instead, with one direct and two indirect pathways, although direct evidence for a second indirect pathway is still lacking. These findings will be of broad interest to neuroscientists across multiple subfields.

---

## [Referee Report · Reviewer #1 (Public Review)]

The manuscript describes an interesting experiment in which an animal had to judge a duration of an interval and press one of two levers depending on the duration. The Authors recorded activity of neurons in key areas of the basal ganglia (SNr and striatum), and noticed that they can be divided into 4 types.

I would like to thank the Authors for performing the analyses I suggested in my previous review - I found their results very interesting and surprising. This is a very interesting and impressive paper.

---

## [Referee Report · Reviewer #2 (Public Review)]

In this valuable manuscript Li & Jin record from the substantial nigra and dorsal striatum to identify subpopulations of neurons with activity that reflects different dynamics during action selection, and then use optogenetics in transgenic mice to selectively inhibit or excite D1- and D2- expressing spiny projection neurons in the striatum, demonstrating a causal role for each in action selection in an opposing manner. They argue that their findings cannot be explained by current models and propose a new 'triple control' model instead, with one direct and two indirect pathways. These findings will be of broad interest to neuroscientists, but lacks some direct evidence for the proposal of the new model.

Overall there are many strengths to this manuscript including the fact that the empirical data in this manuscript is thorough and the experiments are well-designed.

---

## [Author Response]

The following is the authors’ response to the original reviews.

**Reviewer #1 (Public Review):**
The manuscript describes an interesting experiment in which an animal had to judge a duration of an interval and press one of two levers depending on the duration. The Authors recorded activity of neurons in key areas of the basal ganglia (SNr and striatum), and noticed that they can be divided into 4 types.The data presented in the manuscript is very rich and interesting, however, I am not convinced by the interpretation of these data proposed in the paper. The Authors focus on neurons of types 1 & 2 and propose that their difference encodes the choice the animal makes. However, I would like to offer an alternative interpretation of the data. Looking at the description of task and animal movements seen in Figure 1, it seems to me that there are 4 main "actions" the animals may do in the task: press right lever, press left lever, move left, and move right. It seems to me that the 4 neurons authors observed may correspond to these actions, i.e. Figure 1 shows that Type 1 neurons decrease when right level becomes more likely to be correct, so their decrease may correspond to preparation of pressing right lever - they may be releasing this action from inhibition (analogously Type 2 neurons may be related to pressing left lever). Furthermore, comparing animal movements and timing of activity of neurons of type 3 and 4, it seems to me that type 3 neurons decrease when the animal moves left, while type 4 when the animal moves right.I suggest Authors analyse if this interpretation is valid, and if so, revise the interpretation in the paper and the model accordingly.

We thank the reviewer for the general appreciation of the study. Regarding to the interpretation of each SNr subtypes, we have compared firing activities of the same SNr neurons in both standard 2-8 s task and reversed 2-8 s task (Figure 2G-R, Figure S4). Type 1 and Type 2 neurons are related to right and left choices respectively in the standard task (Figure 2G, M, N), and this is even more evident in the reversed 2-8 s task (Figure 2J), because when the movement trajectories of the same mice in 8-s trials were reversed from left-then-right in the control task (Figure 2I) to right-then-left in the reversed task (Figure 2L), the Type 1 SNr neurons which showed monotonic decreasing dynamics in the control 2-8 s task (Figure 2M) reversed their neuronal dynamics to a monotonic increase in the reversed 2-8 s task (Figure 2P). The same reversal of neuronal dynamics was also observed in Type 2 SNr neurons in the reversed version of standard task (Figure 2N vs Figure 2Q). Therefore, Type 1 and Type 2 neurons are related to the action selection. Furthermore, Type 3 and Type 4 SNr neurons exhibiting transient change when mice switching either from left to right, or from left to right maintained the same neuronal dynamics in both standard 2-8 s task and reversed 2-8 s task (Figure S4C-F), indicating that Type 3 and Type 4 neurons are related to the switch between choices but not the specific upcoming choice to be made.

**Reviewer #1 (Recommendations For The Authors):**
Suggest to clarify if SNr neurons recorded just from a single hemisphere or bilaterally.

We have described the recording hemisphere in our Methods (page 46, lines 974-976) as follows “For striatum recording, we implanted 11 mice in the left hemisphere and 8 mice in the right hemisphere. For the SNr recording, we implanted 5 mice in the left hemisphere and 4 mice in the right hemisphere.”

Suggest to analyse if type 1/2/3/4 neurons are preferrably located in hemispheres contra/ipsi lateral to a particular lever or movement.

We have addressed this issue in Figure S3 and Figure S6. In fact, we have implanted electrodes in both left and right hemispheres with mirror M-L coordinates. For striatum recording, we implanted 11 mice in the left hemisphere and 8 mice in the right hemisphere. For the SNr recording, we implanted 5 mice in the left hemisphere and 4 mice in the right hemisphere. We have analyzed the striatal and SNr neuronal activity in left vs. right hemisphere respectively, in relation to action selection. We found that SNr neurons recorded in either left or right hemisphere exhibited the same four types of neural dynamics with similar proportions (Fig. S3). Specially, the Type 1 neurons are dominant in both hemispheres. Similar in striatum, SPNs from left and right hemispheres showed the same four types of neural dynamics with similar proportions (Fig. S6). Therefore, there is no significant difference between hemispheres regarding to the proportion of neuron subtypes.

Suggest to investigate if type 1/2 neurons are involved in preparation for lever press, please investigate if these neurons are also changing their activity during the lever press.

In Figure S1L, we have showed the neuronal activities of example Type 1 and Type 2 SNr neurons to rewarded and non-rewarded lever presses. Type 1 SNr neuron shows higher firing activities when pressing the left lever than pressing the right lever, whereas Type 2 SNr neuron shows higher firing activities when pressing the right lever than pressing the left lever, indicating that Type 1 and Type 2 neurons firing activities are action choice dependent.

Suggest investigating if Type 3/4 neurons are controlling movement from one location to another, please analyse if their activity is correlated with the movement on trial by trial bases.

In Figure S2C-D, we showed firing activities of example Type 3 and Type 4 neurons on trial-by-trial bases. Type 3 neuron showed increased firing activities between 3-4 s during the 8s lever retraction period when the animal switched from left side to right side, whereas Type 4 neuron showed decreased firing activities between 3-4 s during as the animal switching from left to right. We further showed in Figure S4C-F, Type 3 and Type 4 neurons Type 3 and Type 4 neurons are related to the switch between choices but not the specific upcoming choice to be made.

Suggest also performing analogous analyses for striatal neurons.

We showed 4 types of SPNs on the on trial-by-trial bases as follows. Due to the limitation of the number of figures, these data were not included in the manuscript. We have now included these results in Fig. S2(E-H).

Typo: l. 68: "can bidirectionally regulates" -> "can bidirectionally regulate"

Thanks, we have now corrected the typos.

**Reviewer #2 (Public Review):**
In this valuable manuscript Li & Jin record from the substantial nigra and dorsal striatum to identify subpopulations of neurons with activity that reflects different dynamics during action selection, and then use optogenetics in transgenic mice to selectively inhibit or excite D1- and D2- expressing spiny projection neurons in the striatum, demonstrating a causal role for each in action selection in an opposing manner. They argue that their findings cannot be explained by current models and propose a new 'triple control' model instead, with one direct and two indirect pathways. These findings will be of broad interest to neuroscientists, but lacks some direct evidence for the proposal of the new model.Overall there are many strengths to this manuscript including the fact that the empirical data in this manuscript is thorough and the experiments are well-designed. The model is well thought through, but I do have some remaining questions and issues with it.Weaknesses:1. The nature of 'action selection' as described in this manuscript is a bit ambiguous and implies a level of cognition or choice which I'm not sure is there. It's not integral to the understanding of the paper really, but I would have liked to know whether the actions are under goal-directed/habitual or even Pavlovian control. This is not really possible to differentiate with this task as there are a number of Pavlovian cues (e.g. lever retraction interval, house light offset) that could be used to guide behavior.

Sorry for the confusion of task description in the manuscript. We appreciate reviewer’s deep understanding about the complexity of the 2-8 s task we designed. Indeed, the 2-8 s task can’t be simply categorized as goal-directed/habitual or Pavlovian task. There are several behavioral aspects in this task. Lever retraction is served as a Pavlovian cue for mice to start performing the left-then-right sequential movement, but once levers are retracted, there is no cue available to mice during the lever retraction period, and mice have to make a decision to switch choice solely based on its internal estimation of the passage of time, which is considered as a cognitive process. The house light stays on for the entire training session (2 – 3 hours), and will be turned off when the task is done, so house light will not be used as a guidance for choice behavior. The behavior and neural activities during the lever retraction period is our main focus in this manuscript. The main advantage of such task design is that the animal is engaged in a self-determined, dynamic switch of action selection process, which offers a unique opportunity for investigating the role of various neuronal populations in the basal ganglia pathways during action selection.

1. In a similar manner, the part of the striatum that is being targeted (e.g. Figures 4E,I, and N) is dorsal, but is central with regards to the mediolateral extent. We know that the function of different striatal compartments is highly heterogeneous with regards to action selection (e.g. PMID: 16045504, 16153716, 11312310) so it would have been nice to have some data showing how specific these findings are to this particular part of dorsal striatum.

We thank the reviewer for bringing up this point. We are targeting dorsal-central part of striatum. In Figure S5G-L, we showed the specific location we targeted in striatum. Also as specified in Methods (lines 965-970), the craniotomies for electrode implantation were made at the following coordinates: 0.5 mm rostral to bregma and 1.5 mm laterally, and ~ 2.2 mm from the surface of the brain for dorsal striatum. For the virus injection and optic fiber implantation (lines 997-998), the craniotomies was made bilaterally at 0.5 mm rostral to bregma, 2 mm laterally and ~ 2.2 mm from the surface of the brain.

1. I'm not sure how I feel about the diagrams in Figure 4S. In particular, the co-activation model is shown with D2-SPNs represented as a + sign (which is described as "having a facilitatory effect to selection" in the caption), but the co-activation model still suggests that D2-SPNs are largely inhibitory - just of competing actions rather than directly inhibiting actions. Moreover, I am not sure about these diagrams because they appear to show that D2-SPNs far outnumbers D1-SPNs and we know that this isn't the case. I realize the diagrams are not proportionate, but it still looks a bit misrepresented to me.

We appreciate the reviewer’s comments about the diagram. We borrowed and extended the “center-surround” layout from the receptive field of neurons in the early visual system, as an intuitive analogy in describing the functional interaction among striatal pathways (also see Mink 2003 Archives of Neurology). In the co-activation model, if D2-SPNs inhibit the competing action, then the target action will be more likely to be selected due to the reduced competition, which means D2-SPNs actually facilitate the target action in an indirect way. And this is why we define the effect of D2-SPNs in the co-activation model as facilitatory. The area of each region does not represent the amount of cells but mainly qualitative functional role. To make it clearer, we have now added more explanation in the manuscript (page 17, lines 338-341).

4). There are a number of grammatical and syntax errors that made the manuscript difficult to understand in places.

We have now gone through the text carefully and corrected the typos.

1. I wondered if the authors had read PMID: 32001651 and 33215609 which propose a quite different interpretation of direct/indirect pathway neurons in striatum in action selection. I wonder if the authors considered how their findings might fit within this framework.

We appreciate the reviewer’s comments and suggestion. Miriam Matamales et al. (2020, PMID: 32001651) found that dynamic D2- to D1-SPNs transmodulation across the striatum that is necessary for updating previously learned behavior, which highlights the importance of collateral modulations between D1- and D2-SPNs as an additional layer of behavior control besides the classic direct and indirect pathways. This finding is compatible with our “Triple control” model emphasizing the influence of collateral modulations within striatum on behavior choice. James Peak et al. (2020, PMID: 33215609) demonstrated that D2-SPNs are critical to maintain the flexibility of behavior, which is reflected in our “Triple-control” model that activation of D2-SPNs could trigger the behavioral switch from the current action to another action. Although the two studies mentioned above mainly investigate the roles of striatal D1- and D2-SPNs in action learning and behavioral strategies, their functions in general fit within our new ‘Triple-control’ model of basal ganglia pathways for action selection.

1. There is no direct evidence of two indirect pathways, although perhaps this is beyond the scope of the current manuscript and is a prediction for future studies to test.

As accumulating RNA-seq and physiological data implying the heterogeneity of D2-SPNs, the further investigation of the subtypes of D1- and D2-SPNs and their functionality are likely a direction the field will continue to explore. On the other hand, we have discussed other possible anatomical circuits within basal ganglia circuitry that could fulfill the functional role of a third pathway in our new ‘Triple-control’ model, together with or independent of the second indirect pathway (page 32-33, lines 689-700). We certainly hope that our new model will inspire future work to identify and dissect the additional functional pathways in the basal ganglia circuits for action control.

**Reviewer #2 (Recommendations For The Authors):**
Suggestions for authors:1. Consider how specific to the dorso-central striatum these findings are, possibly in the discussion.

We have specified in the Discussion that the study is targeting dorsal-central part of striatum (page 29, lines 609-612).

1. Modify the diagrams in 4S to make them more representative of the model's features.

We have responded this comment above.

1. Consider whether the findings here might fit within the role for direct pathway in excitatory action-outcome learning and the indirect pathway in response flexibility more generally.

The current study is mainly focus on selection and execution of actions. It will definitely be important to continue exploring the functionality of direct vs. indirect pathways in the action learning process.

1. Correct typos and grammatical errors including (but not limited to):a) Line 62-64 - explain why this is controversial? Is it because we don't know which one applies?

In the “Go/No-go” model, indirect pathway inhibits the desired action and function as gain modulation, while in the “Co-activation” model, indirect pathway inhibits the competing action and in turn facilitates the desired action in an indirect manner, therefore these two existing models disagree with each other on the explanation the function of indirect pathway in its targeting action and the net outcome of behavior.

b) Line 68 - Regulates should be regulate.

This has been corrected in the revised manuscript.

c) Line 86 - should read "there are neuronal populations in either the direct or indirect pathway that are activated..."

This has been corrected in the revised manuscript.

d) Line 146-147 - "these types of neuronal dynamics in Snr only appeared in the correct but not incorrect trials" - It seems the authors are suggesting this only for Types 1 and 2 neurons, but this confused me the first time I read it and I suggest it is made clearer.

Line 146-147 now reads “These four types of neuronal dynamics in SNr only appeared…”

e) Line 346 - significant should be significantly.

This has been corrected in the revised manuscript.

f) Line 360 "contrast" should be "contrasting".

This has been corrected in the revised manuscript.